# capTEs enables locus-specific dissection of transcriptional outputs from reference and nonreference transposable elements

Xuemei Li [1,2], Keying Lu[1,2], Xiao Chen[1,2], Kailing Tu[1] & Dan Xie [1✉]

Transposable elements (TEs) serve as both insertional mutagens and regulatory elements in cells, and their aberrant activity is increasingly being revealed to contribute to diseases and cancers. However, measuring the transcriptional consequences of nonreference and young TEs at individual loci remains challenging with current methods, primarily due to technical limitations, including short read lengths generated and insufficient coverage in target regions. Here, we introduce a long-read targeted RNA sequencing method, Cas9-assisted profiling TE expression sequencing (capTEs), for quantitative analysis of transcriptional outputs for individual TEs, including transcribed nonreference insertions, noncanonical transcripts from various transcription patterns and their correlations with expression changes in related genes. This method selectively identified TE-containing transcripts and outputted data with up to 90% TE reads, maintaining a comparable data yield to whole-transcriptome sequencing. We applied capTEs to human cancer cells and found that internal and inserted Alu elements may employ distinct regulatory mechanisms to upregulate gene expression. We expect that capTEs will be a critical tool for advancing our understanding of the biological functions of individual TEs at the locus level, revealing their roles as both mutagens and regulators in biological and pathogenic processes.

[1] Laboratory of Omics Technology and Bioinformatics, Frontiers Science Center for Disease-related Molecular Network, State Key Laboratory of Biotherapy, West China Hospital, Sichuan University, Chengdu, Sichuan 610041, China. [2] These authors contributed equally: Xuemei Li, Keying Lu, Xiao Chen. ✉email: danxie@scu.edu.cn

Transposable elements (TEs) are repetitive DNA sequences that can replicate and insert themselves into other genomic locations[1], constituting approximately half of the human genome[2]. While most TEs have lost their mobilization ability, they can still be actively transcribed in certain contexts, accounting for a substantial fraction of the human transcriptome[3]. A growing body of literature suggests that aberrant TE expression broadly influences human biology[4–6], cancers[7,8] and diseases[9] through retrotransposition and other unknown mechanisms. Furthermore, as TEs are a prolific source of cis-regulatory elements, their insertion can introduce regulatory sequences, potentially affecting the expression of inserted genes[10]. Therefore, measuring transcriptional outcomes resulting from TEs helps to broaden our understanding of how TEs regulate various biological processes.

However, the repetitive nature of TEs and the presence of diverse types of TE-containing transcripts complicate the detection and quantification of TEs at the transcriptional level. The major impediment to next-generation sequencing (NGS)-based strategies comes from the challenge of accurately aligning short reads derived from TEs to the original locus[11]. Despite advances in related algorithms and software that have improved the quantification of TE expression[12–17], the locus-specific analysis of nonreference TEs and young TEs remains challenging due to uncertainties in their transcriptional products and read assignment. The advent of long-read sequencing technologies, such as nanopore sequencing, has provided powerful tools for characterizing TEs[18–20]. Given that less than 10% of reads have TE signals in whole-transcriptome data[21] and that TE-containing transcripts are highly variable, we developed capTEs, a Cas9-based targeted RNA sequencing method on the Oxford Nanopore Technologies (ONT) platform, for specific detection of TE-containing transcripts.

Nanopore sequencing combined with CRISPR/Cas9 technology has been used to study different types of genomic variants[19,22–25], such as structural variations. However, this approach is strongly biased toward detecting DNA strands on one side of Cas9 cleavage sites[19,23,24] and is therefore unsuitable for analyzing both flanking regions simultaneously, which is critical for studying the complete structure of TE-containing transcripts. In addition, the published protocol for this approach performs poorly in terms of both on-target efficiency and data yield, significantly offsetting its enrichment advantages.

The capTEs method overcomes these limitations and efficiently detects both of the flanking sequences of target TEs, with the strand ratio between the two sides of target TEs stabilized at 1.42. Approximately 88% of the output reads contained TEs of interest. Notably, the data yield obtained with capTEs was equivalent to that of regular total RNA-seq (short for nanopore transcriptome sequencing in this study). When combined with a corresponding bioinformatics pipeline, capTEs demonstrated superiority over total RNA-seq in characterizing the transcriptome of internal and inserted nonreference TEs at individual loci and in quantifying the contribution of TEs in various transcription modes to the expression levels of TE-hosting genes. Using the capTEs method, we investigated the functional role of transcribed Alu and LINEs (L1) elements in cancer cells. Our findings indicate that both internal and inserted Alu are implicated in the upregulation of host genes, but potentially through distinct regulatory mechanisms.

## Results

### Development of capTEs. 
We developed capTEs as a strategy for enriching and identifying TE-containing transcripts by selectively detecting the flanking regions of known TEs using Nanopore sequencing (Fig. 1a). In capTEs, we first constructed a full-length cDNA library of total RNA using SMART technology[26,27] for target enrichment. We completely inactivated cDNA ends by dephosphorylation as previously described[23] and the addition of ddNMP to cDNA 3'-ends. We then utilized Cas9-gRNA complexes to create DNA breaks at TEs of interest. By introducing a protease treatment step to digest the Cas9 protein that remains bound to DNA after cleavage[28,29], we ligated sequencing adapters to both ends of the Cas9 cut sites. The protease that we used is a thermolabile enzyme, so no additional purification step was required to remove the protease from subsequent reactions. After sequencing on the Oxford Nanopore Technologies (ONT) platform, we assembled these long reads into transcripts for subsequent analysis (Supplementary Fig. 1d). We first tested and validated this method in K562 cells. The designed gRNAs target Alu and L1 elements, the most active transposons in humans (Supplementary Fig. 1a, Supplementary Table 1). Using the gRNA pool, we successfully captured all expressed Alu and L1 subfamilies detected by total RNA-seq on the ONT platform, identifying a total of 88 branched subfamilies (Supplementary Data 1). Alu and L1 elements were thus considered target TEs in this study.

Previous studies reported a strong strand bias when using nanopore Cas9-assisted targeted sequencing, with over 80% of reads aligned to the Protospacer Adjacent Motif (PAM) side[19,23,24]. In capTEs, the strand ratio between the PAM-distal and PAM sides stabilized at $1.46 \pm 0.20$ (Fig. 1b, c). During the process of adapter ligation, the ligation reaction may occur between the cleaved TE fragments and other cDNAs, resulting in artificial TE-containing transcripts. To assess the side reaction, we spiked synthesized TE cDNA (chimeric sequences of Alu and *E. coli* genomic DNA) into the K562 cDNA library and measured the occurrence of hybrid reads consisting of human cDNAs and the spike-in. The average side reaction rate for capTEs was 0.022, which was slightly lower than the rate of 0.028 for total RNA-seq and threefold lower than the value of 0.066 for the control, in which nanopore-targeted sequencing nCATS[23] was directly applied to capture TE-containing transcripts (Fig. 1d). Notably, the data yield of capTEs reached a comparable level to that of total RNA-seq (Fig. 1e) due to the complete inactivation of DNA ends and removal of Cas9 protein from cut sites. When only the dephosphorylation of DNA ends was performed and the Cas9 protein was not digested after cleavage (control), less than 10% of the pores were reading DNA strands, and adapters took up more than half of the sequencing pores (Fig. 1f). We added ddGMP to the 3'-ends of the DNA molecule to block 3'-hydroxyl residues for adapter ligation, and this virtually eliminated adapter occupancy during sequencing (Supplementary Fig. 1b). To measure the improvement in data yield achieved by digesting the Cas9 protein, we added barcodes to the protease-treated and untreated samples and then pooled them together for library preparation and sequencing. Protease digestion significantly increased the data yield by approximately four times (Supplementary Fig. 1c). With the above optimizations, the capTEs method significantly ameliorated the state of pore occupancy (Fig. 1f) and increased the output data by an average of forty times (Fig. 1e) compared to the control. These results demonstrate the substantial improvements in strand distribution, side-reaction interference and data yields obtained with capTEs compared with existing nanopore-targeted sequencing, indicating the applicability of this method for analyzing TE-containing transcripts.

### Enrichment of target TEs. 
We applied capTEs to three cell lines, including suspended K562 and adherent NCM460 and HCT 116 cells. In successful Cas9-guided enrichment, the reads will start

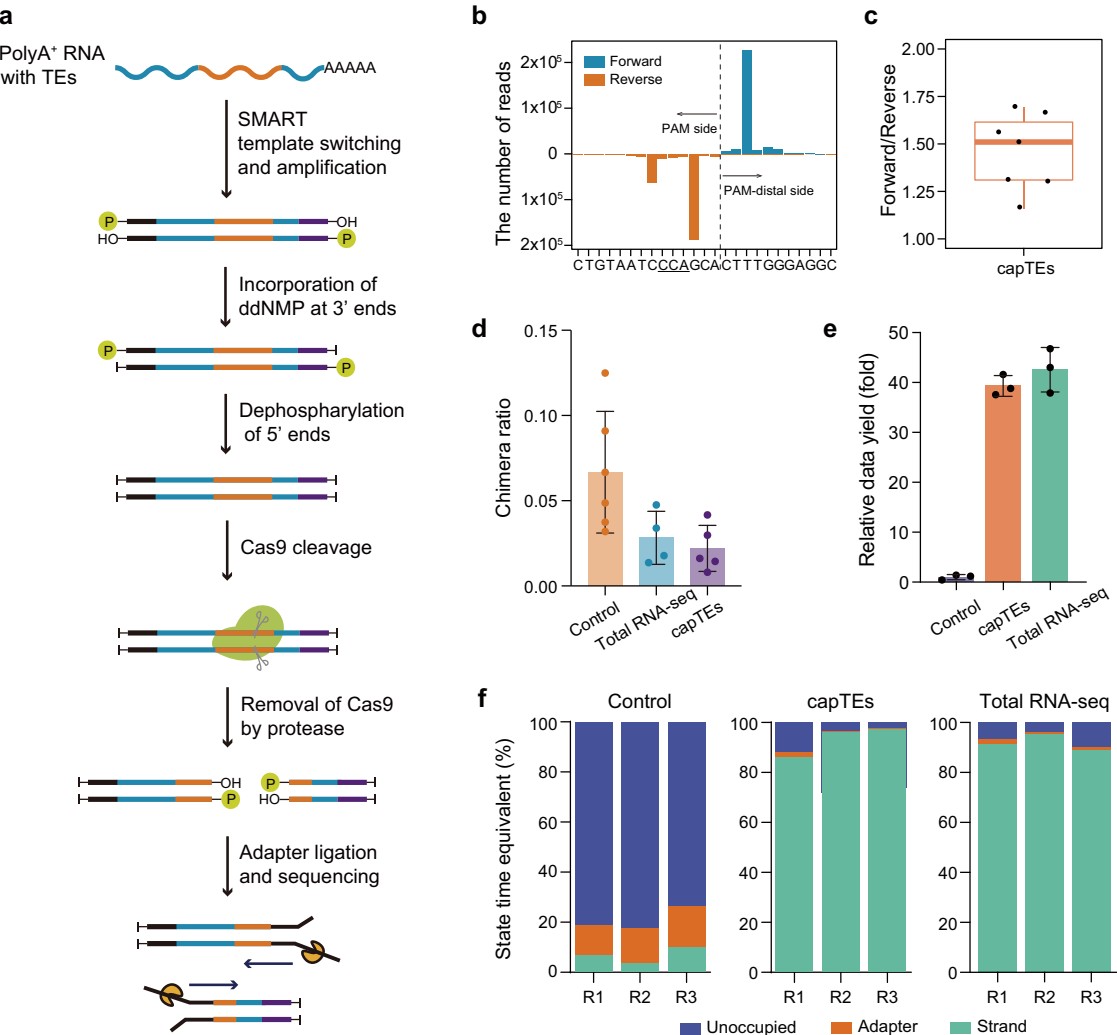

**Fig. 1 Overview of capTEs. a** Schematic of the experimental workflow. The full-length ds-cDNA library was constructed from total RNA usng SMART technology, and cDNA ends were inactivated by ddGMP incorporation to block 3' hydroxyl residues and dephosphorylation to remove 5' phosphate residues. Then, new DNA ends were created by Cas9-gRNAs targeting specific sequences. After the release of Cas9 from the cleavage sites by the thermolabile protease, sequencing adapters were ligated to the cleavage sites for subsequent sequencing. **b** Histogram displaying the strand distribution of capTEs data in the targeted region of Alu gRNA. The x-axis shows the position where the read starts or ends. The underlined uppercase letters represent the PAM sequence, and the dashed line marks the Cas9 cut site. **c** Boxplot showing the strand ratio of capTEs data ($n = 7$). The box edges and whiskers indicate the median, upper and lower quartiles (the 25th and 75th percentiles) and 1.5 × interquartile range, respectively. **d** Bar plots showing side reaction rates of control (orange, $n = 6$), total RNA-seq (blue, $n = 4$) and capTEs (purple, $n = 5$). The side reaction rate is determined by the fraction of hybrid reads among all reads containing spike-in sequences. Error bars represent standard deviation. **e** Bar plot showing the data outputs of capTEs (orange, $n = 3$) and total RNA-seq (green, $n = 3$) relative to the control (gray–purple, $n = 3$), where the control is normalized to 1. Error bars represent standard deviation. **f** Stacked bar plots showing the state composition of available pores in control, capTEs and total RNA-seq: unoccupied pores (blue), adapter-occupied pores (orange) and DNA strand-occupied pores (green). The proportion (y-axis) is determined by the occupied time. **d–f** In the control, nCATS is directly applied to capture TE transcripts.

with the sequences of target TEs. We analyzed the positional distributions of Alu and L1 elements. We found that these TEs were condensed at the heads of the reads (Supplementary Fig. 2). To evaluate the enrichment efficiency, we defined an on-target read as one that contained Alu or L1 within its first 50 nt. The on-target rate was approximately 69% in seven independent tests (Fig. 2a). Approximately 88% of passed reads were available for subsequent analysis because they all contained TEs of interest (Fig. 2a). In our total RNA-seq data, approximately 3% and 10% of reads started with or contained TE signals, respectively (Fig. 2a). Overall, capTEs reached an average of ninefold enrichment for TE-containing reads compared to total RNA-seq. The high on-target rate obtained with capTEs may result from the abundance of TE-containing transcripts.

RNA sequencing is the most commonly used tool for the genome-wide analysis of TE expression. To assess the capability of our developed method for detecting target TEs, we generated 6 Gb of data using capTEs and total RNA-seq methods on the ONT platform and performed a comprehensive comparison between the two methods. The capTEs approach identified a total of 209,646 loci from the Alu and L1 elements, which was five times greater than the 30,189 identifications obtained from total RNA-seq (Fig. 2b). Considering all the Alu and L1 loci identified by both methods (212,902 loci), capTEs achieved a detection rate of 98.5%, leaving only 3256 loci undetected (Fig. 2b). Likewise, capTEs outperformed total RNA-seq in identifying young TE sites (defined as those less than two million years old). It captured 97% of the total number of young TE loci detected by both

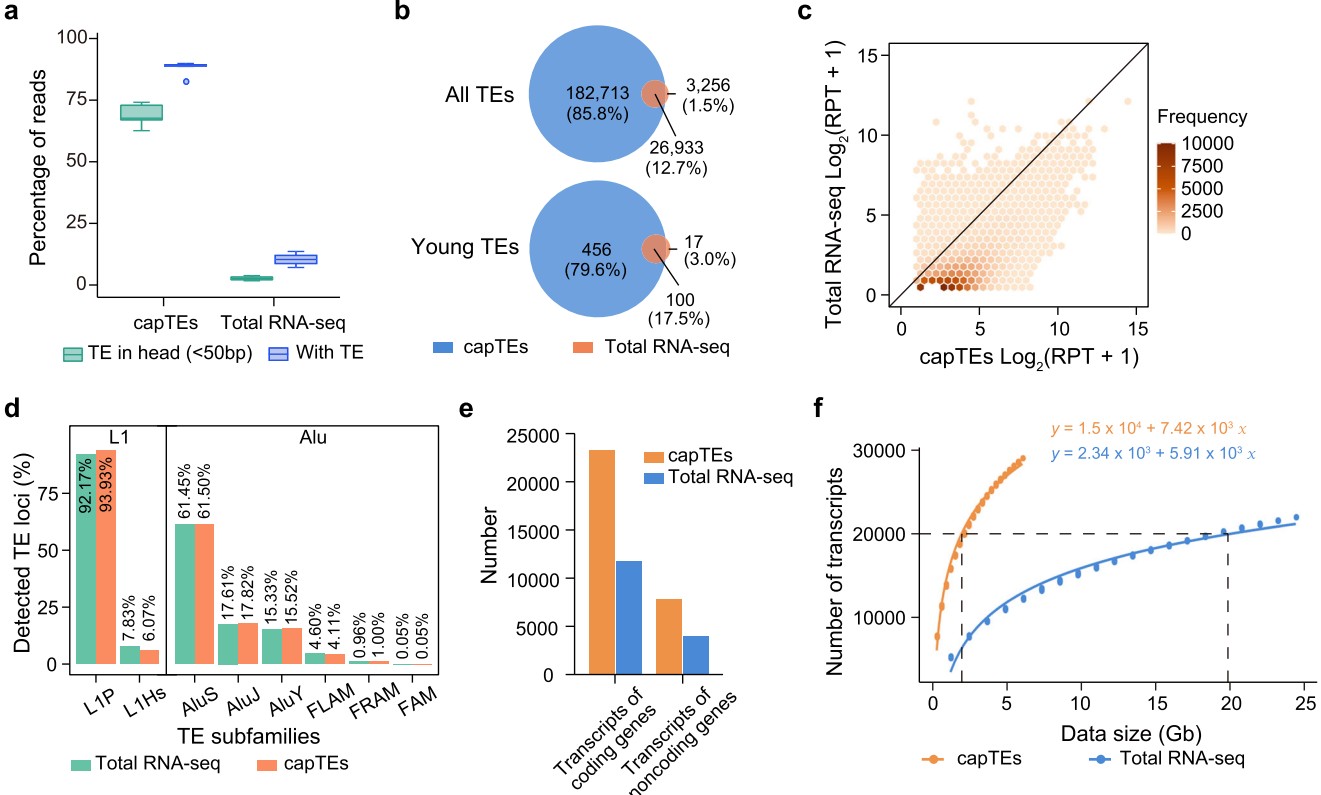

**Fig. 2 Enrichment efficiency of capTEs. a** Box plot showing the percentage of on-target reads (green, reads containing TEs of interest at the beginning 50 nt) and reads containing TEs of interest (blue) among all reads generated by capTEs ($n = 7$) and total RNA-seq ($n = 3$). The box edges and whiskers indicate the median, upper and lower quartiles (the 25th and 75th percentiles) and 1.5 × interquartile range, respectively. **b** Venn diagram showing the overlap between the target TE and young TE loci detected by capTEs and total RNA-seq methods. **c** Dot plot showing the coverage of target TEs in capTEs (x-axis) and total RNA-seq (y-axis) data. TEs detected by both capTEs and total RNA-seq are included in the analysis. RPT represents reads per TE. **d** Bar plots showing the subfamily proportions of target TEs detected by capTEs (orange) and total RNA-seq (green). **e** Bar plot showing the number of TE-containing transcripts identified with 6 Gb of capTEs (orange) and total RNA-seq (blue) data. **f** Saturation curves of TE transcript identifications using capTEs (orange) and total RNA-seq (blue). Black dashed lines indicate data requirements for 20,000 identifications.

methods in Alu and L1 elements, which was five times higher than the detection rate achieved by total RNA-seq (Fig. 2b). Moreover, capTEs exhibited higher coverage for the detected target TEs (Fig. 2c). Genomic DNA would interfere with the detection of transcribed TEs, especially when using targeted sequencing. To examine whether the efficient detection of target TEs using our method was a result of genomic DNA contamination, we included control samples that were not subjected to reverse transcription. In these samples, we did not detect TE signals, which strengthens the validity of our findings. To investigate whether capTEs exhibited any bias toward some specific subfamilies of Alu and L1 elements, we conducted a comparative analysis of TE subfamily proportions within the target TE loci identified by capTEs and total RNA-seq. Our analysis demonstrated no significant difference in the overall proportions of TE subfamilies between the two methods (Fig. 2d, Supplementary Table 2). The maximum enriched fold observed in the capTEs dataset (the ratio between the subfamily proportion of capTEs and total RNA-seq) was limited to 1.2 (Supplementary Table 2). These results indicate that our method effectively and modestly detects various subfamily members of Alu and L1 elements without displaying a substantial bias toward specific subfamilies. We also assessed the ability of capTEs to detect target TEs at the transcript level. By aligning Alu- and L1-containing reads to the Gencode annotated transcripts, we identified 31,395 matched transcripts using capTEs, which is twice the number detected using total RNA-seq (Fig. 2e). Saturation analysis

revealed that total RNA-seq would require 137 Gb of data to achieve the same level of identification (Fig. 2f).

Taken together, these comparisons demonstrate the advantages of our method over total RNA-seq in terms of enriching and identifying target TEs, including young TE loci within their respective families, with fewer data requirements but more identifications.

**Identification of noncanonical TE transcripts and transcribed nonreference TEs.** To estimate the feasibility of reconstructing TE-containing transcripts with our method, we assembled transcripts using K562 data generated by capTEs, nCATS (control) and total RNA-seq and analyzed the completeness of transcripts that were identifiable in Gencode annotations. We found that 36% of the transcripts assembled from nCATS data were embedded in canonical transcripts, most likely due to strand bias (Supplementary Fig. 3a,b). In comparison, the assembly of capTEs data contained a higher percentage of full-length transcripts and was similar in length to that of total RNA-seq data (Supplementary Fig. 3b,c). This suggests that it is acceptable to assemble TE-containing transcripts with capTEs data.

We next investigated noncanonical transcripts. We identified 18,298 noncanonical transcripts with capTEs data, 98.4% of which overlapped with at least one TE of interest, confirming the high specificity of our method (Supplementary Fig. 4a, Supplementary Data 2). Consistent with the integrity assessment of assembled canonical transcripts, the noncanonical transcripts

COMMUNICATIONS BIOLOGY | (2023)6:974 | https://doi.org/10.1038/s42003-023-05349-1 | www.nature.com/commsbio

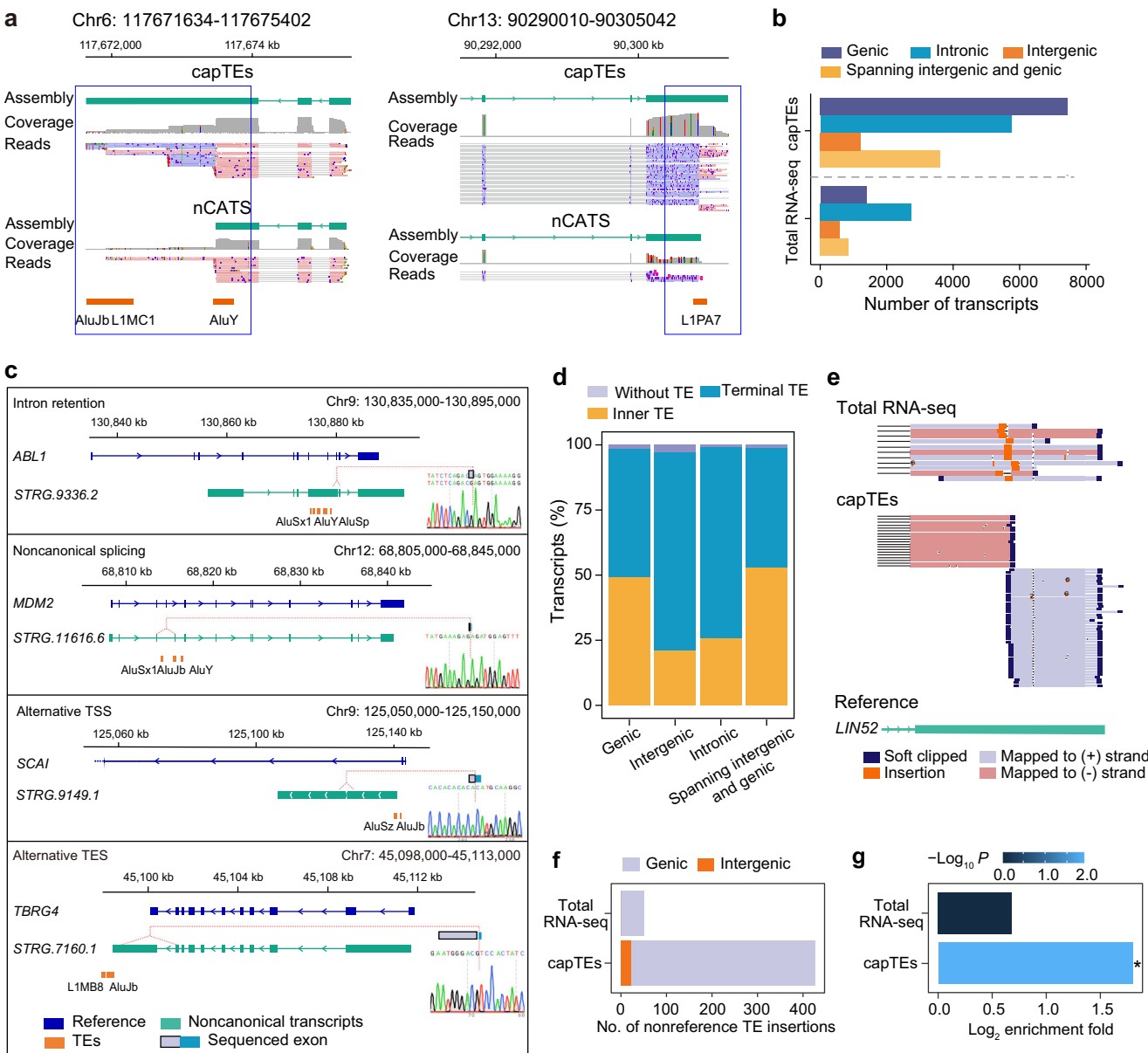

**Fig. 3 Noncanonical transcripts and transcribed TE insertions identified with capTEs. a** Genome browser view showing two examples of noncanonical transcript assembly for capTEs and nCATS data. The blue rectangle marks the incomplete assembly of nCATS data. **b** Bar plots showing the number of noncanonical TE-containing transcripts detected by capTEs and total RNA-seq in various genomic regions. **c** Genome browser view and Sanger sequencing validation of each transcription pattern for TEs to generate noncanonical transcripts. The nucleotide signal displays the splicing junction. **d** Stacked bar plots showing the percentages of noncanonical transcripts with inner TEs (yellow), with terminal TEs (blue) and without TEs (gray) out of the various types of noncanonical transcripts identified by capTEs. Alu and L1 represent TEs targeted by designed gRNAs. **e** Genome browser view of TE insertions identified with capTEs and total RNA-seq. **f** Stacked bar plot showing the number of TE insertions identified by capTEs and total RNA-seq in the genic (purple) and intergenic (orange) regions. **g** Enrichment fold values of TE insertions in oncogenes. Significant * represents BH-adjusted $P$ value < 0.05 reported by the Chi-squared test between insertions and all expressed TEs.

identified with capTEs were longer than those identified by nCATS (Fig. 3a). According to the genomic context, we classified noncanonical TE-containing transcripts into the following four categories: genic (transcripts in gene regions but not fully contained within introns), intronic (transcripts fully contained in the intronic interval), intergenic (transcripts fully contained in intergenic regions) and those spanning intergenic and genic. Overall, our method identified more noncanonical transcripts of all types than total RNA-seq (Fig. 3b). The majority of the identified noncanonical transcripts overlapped with genic regions (Fig. 3b).

To characterize the effects of target TEs on the production of noncanonical transcripts, we analyzed the transcription patterns of noncanonical TE-containing transcripts and validated them by PCR followed by Sanger sequencing (Fig. 3c). Alu sometimes appears in mature mRNAs through alternative splicing[30,31]. Here, we reported 7566 splicing-mediated noncanonical transcripts, 4706 and 2,860 of which contained intron retention and noncanonical splicing junctions, respectively (Supplementary Data 2). TEs play a substantial role in the generation of intronic and intergenic noncanonical transcripts by serving as alternative transcription start or end sites (TSSs and TESs), especially in

cancer cells where the global loss of DNA methylation occurs in repeated DNA regions[32,33]. Consistent with this knowledge, over 70% of intergenic and intronic transcripts identified by total RNA-seq overlapped with Alu or L1 elements (Supplementary Fig. 4b). With capTEs, we identified 7,068 transcripts in intronic or intergenic regions (Fig. 3b), of which more than 70% overlapped TEs at TSSs or TESs (Fig. 3d).

TE insertions have attracted much attention in the etiologic studies of different cancers due to their potential to disrupt gene functions[34–38]. Thus, there is a great need to identify insertions at the transcriptional level. Considering that most capTEs reads started with TEs that would be soft-clipped during the mapping process (Fig. 3e), we applied PALMER[19,39], a nonassembly based tool, to call TE insertions. A total of 431 and 53 insertions were identified in the capTEs and total RNA-seq data under a certain threshold of supporting reads (Fig. 3f, Supplementary Data 3). We next examined whether these nonreference TEs detected by capTEs existed in the K562 genome. We performed the same analysis of 30× nanopore whole genome sequencing (WGS) data of K562 cells, and 50% of insertions were identifiable with at least one supporting read. The validation rate was likely due to our extensive identification of transcribed insertions that were missed by the WGS method, as we found only 81 insertions in the transcribed regions using the WGS data under the same threshold as capTEs (Supplementary Data 3). By analyzing the genomic context, we found that 95% of insertions identified by capTEs were located in intragenic regions and concentrated in oncogenes (Fig. 3f, g), implying the potential for the application of our approach to discover the causes of cancers and other diseases.

In summary, these results showed the application of capTEs in revealing various types of TE-containing transcripts and transcribed TE insertions. This enabled us to explore the complex transcription patterns and genomic context of expressed reference and nonreference TEs.

**Locus-specific quantification of TE expression**. Short read lengths pose many challenges for measuring TE expression at a single-locus resolution. Since our data were long-read data, we attempted to extend their use for quantifying TE expression. We performed capTEs and quantitative analyses on the expression levels of target TEs, TE-containing transcripts, and their host genes in three biological replicates of the breast cancer cell line MDA-MB-231 and the non-tumorigenic cell line MCF 10 A (Supplementary Data 4). To test the reliability of the quantitative results, we first validated the expression levels of transcripts containing target TEs, specifically Alu and L1 elements. The Pearson correlations between biological repeats exceeded 0.97, which demonstrated the high reproducibility of our method (Supplementary Fig. 5a). When comparing our results to those determined from NGS data[40], we observed good agreement for the expression levels of TE-containing transcripts (Supplementary Fig. 5b). We further carried out qPCR validation of 25 randomly selected TE-hosting genes, which were identified to host TE-containing transcripts (see Methods). The qPCR results were consistent with the quantitative results of capTEs (Supplementary Fig. 5c). These analyses demonstrate the feasibility of capTEs to measure the expression levels of genes and transcripts containing target TEs.

The assignment of multimapped reads has been the major issue in quantifying TE RNA abundance. The unique mapping rate of capTEs reads was approximately 80% (Fig. 4a), similar to that of long-read total RNA-seq[41]. Therefore, we attempted to interrogate the expression levels of TEs in the same way as we analyzed genes. To evaluate the accuracy, we counted reads from synthetic TE cDNAs that were mixed into the cDNA libraries in a known quantity. We observed high consistency between the incorporated number of molecules and read counts for the spike-in by linear regression (Fig. 4b), indicating the feasibility of estimating TE expression levels with capTEs data. We next compared our TE expression results to those obtained from NGS data using two independent TE-dedicated bioinformatics tools, TEtranscripts[12] and Telescope[17]. Our analysis revealed that capTEs detected expression changes at 65,631 Alu and L1 loci between MDA-MB-231 and MCF 10 A cells, while TEtranscripts and Telescope quantified 25,383 and 19,124 loci, respectively, based on NGS data (Fig. 4c). When examining the subfamily proportions, we observed a substantially higher representation of evolutionarily young TE subfamilies, specifically Alu Y and L1Hs, within the target TE loci quantified by capTEs compared to NGS-based methods (Fig. 4d, Supplementary Table 3). The proportions of Alu Y and L1Hs in capTEs-quantified target TEs were approximately 1.8 times and 17.6 times, respectively, of those observed in the NGS results (Fig. 4d, Supplementary Table 3). Moreover, our analysis of young TE loci revealed that their proportion within the Alu and L1 loci detected by capTEs was on average 5.8 times higher compared to NGS-based methods (5-fold higher than TEtranscripts and 6.5-fold higher than Telescope, Fig. 4e). These results highlight the superiority of capTEs over NGS in quantifying the expression of young TEs, which are known to exhibit higher sequence similarity than old TEs. We suspect that this advantage may be attributed to the long-read nature of our capTEs data. We further compared our quantification results to those reported by Telescope, as this tool was specifically designed for the locus-specific analysis of TE expression. Two pieces of evidence suggested that capTEs presented a higher sensitivity than NGS in detecting modest changes in the expression of individual TEs. First, compared to NGS, capTEs identified more differentially expressed TEs at various evolutionary ages, and these differential TE sites accounted for a greater proportion of all quantified target TE sites (Fig. 4f, Supplementary Fig. 6a). Second, we observed lower expression changes for differential TEs detected only by capTEs in comparison to those revealed by both capTEs and NGS (Supplementary Fig. 6b).

A previous study reported the overexpression of young TEs in cancers, possibly due to the loss of DNA methylation in surrounding genomic regions[16]. However, the significantly upregulated Alu and L1 elements that we detected in breast cancer cells did not show significant enrichment in any subfamilies (Fig. 4g). Considering that retrotransposons can initiate transcription from their own promoters, we speculated that young TEs tend to be active in the mode of autonomous transcription. Based on the details of the assembled TE-containing transcripts (Supplementary Data 5), we examined the autonomous transcription levels of individual TEs according to the abundance of noncanonical transcripts starting with the individual TE (Fig. 4h). In this analysis, we identified 2,630 independently transcribed Alu and L1 loci in MDA-MB-231 breast cancer cells (Supplementary Data 5), most of which were cell specific. We further characterized the ratio of autonomous transcription at each locus in driving transcription (i.e., the autonomous transcription level of the given locus relative to the additive abundance of all transcripts containing that TE) (Fig. 4i). As expected, we observed a significant enrichment of the young subfamily, AluY, in fully autonomous loci (Fig. 4j).

Together, these results showed advances in the locus-specific measurement of expression for target TEs, including young TEs, through the utilization of the capTEs method. Additionally, this method facilitates the investigation of autonomous transcription at specific loci.

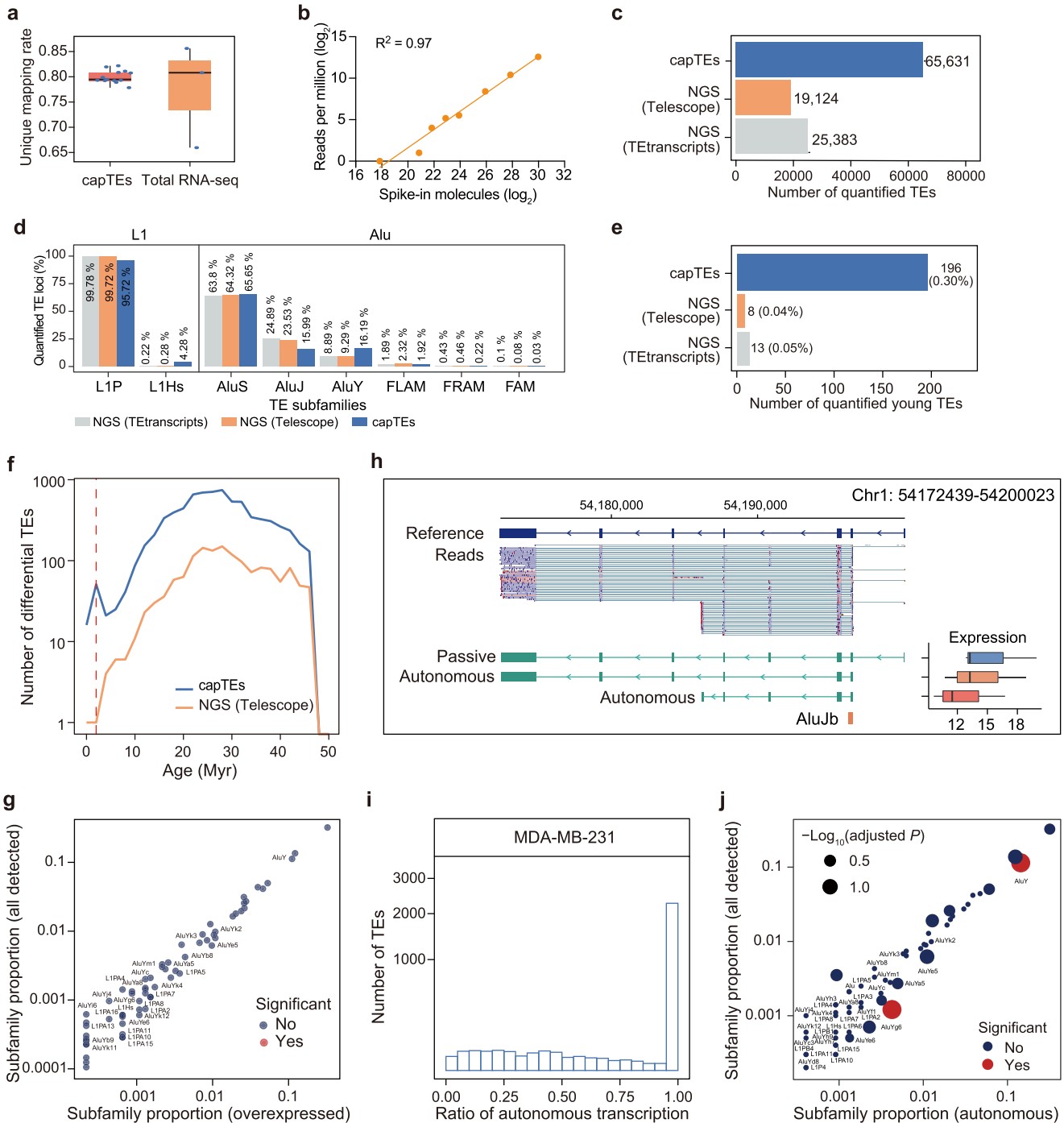

**Characterizing the impacts of internal and inserted TEs on the transcriptome of cancer cells**. Providing comprehensive information on the expression of TEs and their host genes, capTEs allowed us to characterize the transcriptional contribution of target TEs. We profiled the transcriptome of Alu and L1 elements using capTEs in breast and colorectal cancers and matched normal cells. This profiling included the measurement of expression levels for each Alu and L1 locus and its host gene, as well as the assembly of noncanonical transcripts containing target TEs (Supplementary Data 4–6). The transcriptional alterations of differentially expressed target TEs were highly consistent with those of their host genes in cancer cells, as demonstrated by a Spearman correlation coefficient of 0.94 (Fig. 5a, Supplementary Fig. 7a). By quantifying the number of noncanonical transcripts

under each TE-hosting gene, we observed a positive correlation between the relative number of noncanonical transcripts (calculated as the difference in noncanonical transcript counts between cancer cells and control cells) and the expression changes of the TE-hosting genes in cancer cells (Fig. 5a, Supplementary Fig. 7a). This suggests that the production of noncanonical transcripts may be one of the mechanisms underlying the upregulation of TE-hosting genes. To explore this possibility further, we examined the abundance of canonical and noncanonical transcripts and found that noncanonical transcripts were substantial contributors to gene-level alterations (Fig. 5b). For example, we confirmed by qPCR that the overexpression of *FOXRED2* in breast cancer cells was due to the overexpression of noncanonical transcripts rather than the canonical transcripts (Fig. 5c). To

**Fig. 4 Assessing the ability of capTEs to the locus-specific measurement of TE expression. a** Boxplot showing the unique mapping rate of capTEs ($n = 13$) and long-read total RNA-seq ($n = 3$) data. **b** Scatter plot showing the correlation between the number of incorporated TE cDNA molecules (x-axis) and the number of reads detected with capTEs (y-axis). **c** Bar plot showing the number of TEs quantified with capTEs (blue), Telescope (orange) and TEtranscripts (gray). **d** Bar plots showing the subfamily proportions of target TEs detected by capTEs (blue), Telescope (orange) and TEtranscripts (gray). **e** Bar plot showing the number of young TEs quantified with capTEs (blue), Telescope (orange) and TEtranscripts (gray). The percentages within the parentheses represent the proportion of young TEs among all the target TEs quantified by the respective method. **f** Line chart showing the number of differential target TEs at various evolutionary ages identified with capTEs and Telescope. The red dashed line represents the cutoff for young TEs, which is set at 2 million years (Myr). **g** Scatter plots show the proportion of each TE subfamily among overexpressed (x-axis) and all detected target TEs (y-axis). The P value was determined by Fisher's exact test between overexpressed target TEs and all expressed target TEs, and a significant change (solid red circle) was defined as BH-adjusted $P < 0.05$. **h** Genome browser view showing an example of measuring autonomous transcription levels of TEs at specific loci. The boxplot displays the expression levels of assembled transcripts and the colors indicate the transcripts where the analyzed TE was in autonomous (red), autonomous (orange) and passive (blue) transcription modes. The autonomous transcription level of this TE locus is represented by the total expression levels of the two transcripts labeled as "autonomous". The expression levels are indicated by normalized read counts. **i** Histogram showing the number of TEs at various degrees of the independent promotion of transcription in breast cancer cells, ranging from 0 (passive) to 1 (fully autonomous). Passive transcribed TEs are not counted. **j** Scatter plots show the proportion of each TE subfamily in fully autonomously transcribed (x-axis) and all detected target TEs (y-axis). The P value was determined by Chi-squared test between fully autonomously transcribed target TEs and all expressed target TEs, and significant change (solid red circle) was defined as BH-adjusted $P < 0.05$. **c–f** NGS data were analyzed using Telescope and TEtranscripts. **a, h** In boxplots, the box edges and whiskers indicate the median, upper and lower quartiles (the 25th and 75th percentiles) and 1.5 × interquartile range, respectively.

determine whether differentially expressed TEs are involved in generating noncanonical transcripts, we conducted an analysis of the expression levels of differential TEs and their corresponding TE-derived transcripts that began or ended with the TE in question. As expected, we observed that the density of TE-derived transcripts changed in concordance with the expression levels of TEs, with higher TE expression levels corresponding to higher transcript densities (Fig. 5d). These results indicate a role of TEs in regulating gene expression.

To investigate how individual TEs contribute to the changes in the expression of their host genes, we divided the transcription of each overexpressed TE into three modes based on the transcription patterns of the transcripts containing that TE: autonomous transcription, intron retention, and passive transcription. We then calculated the proportion of transcript abundance for each transcription mode, which we used as the transcriptional contribution of that particular mode. Our analysis revealed that, consistent with a previous study[11], the majority of TE RNAs originated from the promoter activity of host genes. This was supported by the observation that the upregulated TE loci were predominantly transcribed through intron retention (14%) and passive transcription (65%) (Fig. 5e). Furthermore, we found that 19% of TEs were transcribed via more than one transcription mode (Fig. 5e). Interestingly, we identified over 200 over-expressed TE loci where transcription was initiated independently, and some of them were located within the gene bodies of cancer-related genes. For example, AluY (chr19: 4099969-4100275) and L1PA4 (chr14: 61707309-61708779) initiated transcription in the genomic regions of *MAP2K2* (Mitogen-activated protein kinase kinase 2) and *HIF1A* (hypoxia-inducible factor 1 subunit alpha), respectively, and generated noncanonical transcripts (Fig. 5f). These transcripts accounted for approximately 49% and 50% of the total expression levels of their host genes, *MAP2K2* and *HIF1A*, respectively (Fig. 5e). However, the TE-initiated transcripts had nonidentical sequences compared to the classical transcripts (Fig. 5f); thus, their functions require further investigation.

We next investigated whether nonreference TEs shared similar transcriptional characteristics with internal TEs in cancer cells. Using capTEs, we identified 777 and 459 transcribed Alu and L1 insertions in breast and colorectal cancer cells, respectively (Supplementary Fig. 7b, Supplementary Data 7). Consistent with our observation in K562 cells, the genomic locations of these insertions were significantly condensed in oncogenes compared

to all expressed internal TEs (Supplementary Fig. 7c), suggesting potential cancer risks associated with retrotransposition events. We found that the transcriptional events of nonreference TEs were positively correlated with the expression changes of their inserted genes in cancer cells (Fig. 5g), similar to upregulated internal TEs. To gain further insight, we analyzed the genomic context of inserted and internal TEs by families, revealing the difference between nonreference and reference Alu elements. The density of inserted Alu elements peaked in TESs, while that of expressed internal Alu elements plateaued across the gene body (Fig. 5h). Furthermore, we observed that the genomic locations of Alu insertions displayed a significant preference for 3'UTRs, which are known to be enriched in expression quantitative trait loci (eQTL)[42], compared to the background (expressed Alu) (Supplementary Fig. 7d, Fig. 5i). To determine whether this distribution bias was due to specific Alu subfamilies instead of insertions, we further analyzed the genomic context of expressed Alu subfamilies, including Alu Y, Alu J, and Alu S. However, we did not detect significant differences in the proportion of 3'UTR between these subfamilies and the Alu family as a whole, and we repeatedly observed a distribution preference of inserted Alu for the 3'UTR compared to each of the Alu subfamilies described above (Fig. 5i, Supplementary Fig. 7d), indicating that enrichment in the 3'UTR is a specific feature of expressed inserted Alu. Taken together, these results suggest a potential mechanism by which Alu insertions act in 3'UTRs to regulate the expression of their inserted genes. This is consistent with previous reports that some Alu insertions can provide eQTLs[43] and alter gene expression[44].

## Discussion

Here, we developed a long-read targeted RNA sequencing tool, capTEs, that can provide a comprehensive profile of TE-associated transcriptomes. After extensive optimization of the enrichment protocol, capTEs achieved a substantial improvement in targeting efficiency and data yield. Unlike existing Cas9-based nanopore sequencing targeting methods, which are biased for the single-sided detection of target sites[19,23,24], capTEs detects both flanking regions of the given sequences, enabling the complete assembly of TE-containing transcripts. In addition, capTEs is applicable to multiplexing samples and requires merely 2 μg of input cDNA in total, indicating its potential applicability in single-cell investigations.

An increasing number of studies have emphasized the critical role of TE expression in biological processes. However, owing to

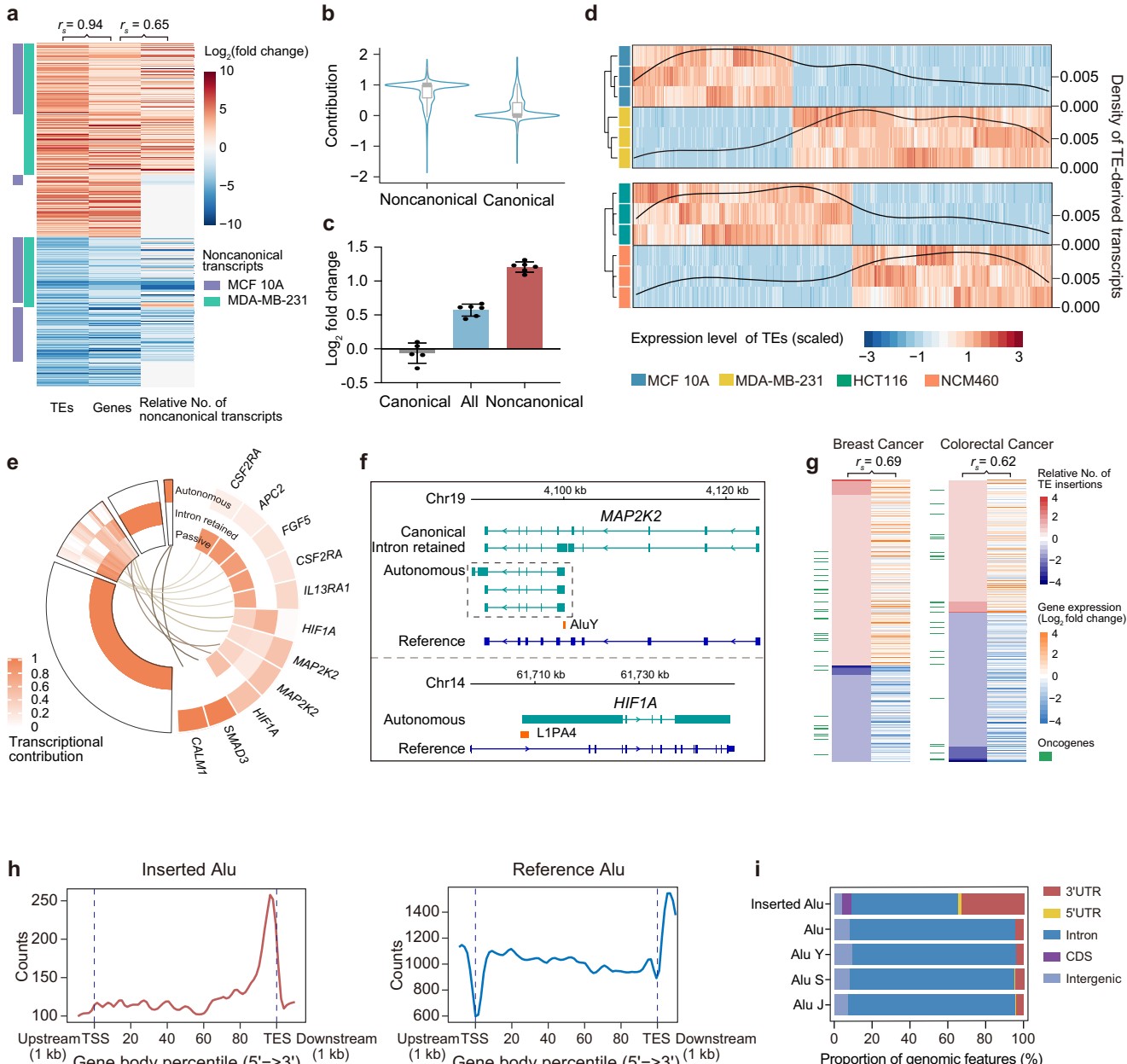

**Fig. 5 Transcriptional changes related to target TEs in cancer cells. a** Heatmap depicting the correlation among expression changes of target TEs, expression changes of TE-hosting genes and relative number of noncanonical transcripts in MDA-MB-231 cells compared to MCF 10 A cells. $r_s$ represents Spearman's correlation coefficient. **b** Violin plot showing contributions of noncanonical transcripts (noncanonical) and canonical transcripts (reference) to the changes in the expression of TE-hosting genes in MDA-MB-231 cells compared to MCF 10 A cells. The contribution is the ratio of expression changes of the noncanonical or reference transcript to that of all transcripts in each locus. In the boxplot, the box edges and whiskers indicate the median, upper and lower quartiles (the 25th and 75th percentiles) and 1.5 × interquartile range, respectively. **c** Bar plot showing qPCR measurements of expression changes in *FOXRED2* reference transcripts (gray, $n = 6$), noncanonical transcripts (red, $n = 6$) and all transcripts (blue, $n = 6$) in MDA-MB-231 cells compared to MCF 10 A cells. Error bars represent standard deviation. **d** Heatmaps depicting the expression levels of differential TEs in MDA-MB-231 and HCT 116 cells compared to their matched normal cells ($n = 3$), and density lines showing the frequency of TE-derived transcripts at each TE locus. **e** Circular heatmap showing following three-category contribution of each overexpressed TE loci to the expression levels of its host genes, autonomous transcription, intron retention and other passive transcription in MDA-MB-231 cells. Each ring represents one type of contribution. The enlarged portion (right) shows autonomously transcribed TEs that overlap with genes involved in cancer pathways. **f** Genome browser view showing examples of noncanonical TE-containing transcripts identified in MDA-MB-231 cells within cancer pathway genes, *MAP2K2* and *HIF1A*. **g** Heatmap showing the relative frequency of TE insertions in cancer cells compared to matched normal cells (left) and transcriptional changes in TE-inserted genes (right). The color bar displays insertions in oncogenes (green). $r_s$ (Spearman's correlation coefficient) between the relative frequency of TE insertions and changes in gene expression are 0.69 for MDA-MB-231 cells and 0.62 for HCT 116 cells. **h** Distribution of expressed inserted Alu (red) and expressed reference Alu (blue) in gene bodies. **i** Stacked bar plots showing the proportion of genomic features (3'UTR, 5'UTR, intron, CDS and intergenic regions) in expressed inserted Alu and expressed reference Alu, Alu Y, Alu S and Alu J.

the limitation imposed by short reads, it is still challenging to systematically characterize TE-associated transcriptomes via NGS-based methods[11]. The long-read and high-depth data generated by capTEs allowed the broad elucidation of TE-containing transcripts derived from intron retention, autonomous transcription, and other processes, and precise assignment of TE reads to their original loci, which enabled us to determine TE expression in various transcription patterns. By applying capTEs to cancer cells, we identified autonomous Alu and L1 loci that contributed to the overexpression of genes involved in cancer pathways. Unlike internal Alu elements, the nonreference Alu elements were significantly enriched in 3'UTRs, although they were also positively correlated with the expression of their inserted genes. This case study demonstrates the application of capTEs in investigating the putative regulatory roles of reference and nonreference TEs, which lays a foundation for further mechanistic studies.

Retrotransposon-mediated insertions, which occur at a high frequency in cancers, pose potential risks for carcinogenesis and pathogenesis. Existing tool primarily designed for WGS data[45] often identify TE insertions in the genome which contain a large proportion of neutral mutations. In contrast, capTEs specifically focuses on transcribable TE insertions that lead to noncanonical transcripts, which are likely to have biological functions. As expected, capTEs identified a substantial number of transcribed TE insertions that were missed by long-read WGS or total RNA-seq methods. Notably, these TE insertions were found to occur more frequently in oncogenes and exhibit a strong correlation with gene expression. This highlights the importance of studying transcribable TE insertions to explore their potential regulatory roles in cancer and other biological processes.

The capTEs method has been validated based on two distinct TE families, Alu and L1 elements, which have very different genomic distributions. However, to apply this method to other types of TEs, we suggest the application of several tests to ensure proper execution and feasibility, including tests of gRNA cutting efficiency, the positional distribution of targets in reads, and strand bias. In theory, if targeted enrichment is successful, the target sequence should be concentrated at the beginning and ends of the reads. Although our method considerably improves strand bias, we found that the nucleotide sequence at the cutting site of Cas9-gRNA complexes may affect the strand distribution to some extent, as observed in our preliminary testing of different gRNAs. Furthermore, if quantitative analysis of TE expression is desired, we recommend incorporating a standard containing known concentrations of the target sequence for quality control. Currently, our method produces data with an approximately 80% unique mapping rate, comparable to total RNA-seq data produced on the Nanopore sequencing platform. However, accurately quantifying TE expression levels requires further improvement, which is expected to be achieved by introducing UMIs and enhancing sequencing accuracy.

In summary, the capTEs method has been developed to generate long-read, high-coverage, and quantifiable data for transcribed reference and nonreference TEs, enabling the quantitative analysis of TEs in diverse transcription modes at individual loci. The high on-target rate significantly reduces the demand for data size, making this method suitable for pan-cancer analysis and population studies of various diseases and phenotypes.

## Methods

**Cell culture**. All cell lines were grown in 6 cm dishes at 37 °C in a 5% $CO_2$ incubator. The K562, MDA-MB-231 and HCT 116 cell lines were cultured in high-glucose DMEM supplemented with 10% fetal bovine serum and 1% penicillin-streptomycin

antibiotics (pen-strep). NCM460 cells were cultured in RPMI 1640 medium supplemented with 10% fetal bovine serum and 1% pen-strep. MCF-10A cells were cultured in DMEM/F12 medium (Ek-Bioscience) containing 5% horse serum, 1% pen-strep, 20 ng/ml EGF, insulin, hydrocortisone and cholera toxin.

NCM460 were obtained from INCELL, and K562, MDA-MB-231, MCF-10A and HCT 116 cells were obtained from ATCC (American Type Culture Collection). All cells were subjected to authentication through short tandem repeat (STR) profiling to confirm their identity and purity. Additionally, regular testing confirmed the absence of mycoplasma contamination in these cell lines.

**RNA extraction and full-length cDNA library construction**. Total RNA was extracted using TRIzol reagent (Invitrogen, 15596026) according to the manufacturer's instructions. Briefly, approximately $2 \times 10^6$ cells were lysed in 1 ml of TRIzol at room temperature for 5 min to ensure thorough cell lysis. Subsequently, 200 µL of chloroform was added to the lysate, followed by centrifugation to separate the RNA-containing aqueous phase. The RNA was then precipitated by adding 500 µL of isopropanol, and the resulting RNA pellet was dissolved in nuclease-free water to obtain the final RNA sample. The RNA concentration was determined using Qubit RNA HS Assay Kit (Thermo Fisher Scientific, Q32852), and the purity and integrity were assessed by NanoDrop One, Qubit and agarose gel electrophoresis. If the RNA is contaminated with genomic DNA, we recommend DNase digestion. The full-length cDNA library was prepared according to the SMART[26,27] protocol. Briefly, reverse transcription was performed by sequentially mixing 2 µg of total RNA with 2 µL of 10 µM oligo-dT primer, 2 µL of 10 mM dNTP mix (New England Biolabs, N0447), 4 µL of 5× RT Buffer, 40 U of RNaseOUT (Invitrogen, 10777019), 2 µL of 10 µM template-switching oligos and 200 U of Maxima H-minus reverse transcriptase (Thermo Scientific, EP0751) in a total volume of 20 µL. Before the addition of RT Buffer, the RNA was denatured at 65 °C for 5 min and immediately chilled on ice. The reverse transcription program was as follows: 42 °C for 90 min, followed by 10 cycles of 50 °C for 2 min and 42 °C for 2 min. Then, a third of the volume of the cDNA product was mixed with VAHTS HiFi Amplification Mix (Vazyme, N616) and 0.5 µM (final concentration) oligos for PCR amplification. The mixture was incubated at 98 °C for 3 min, followed by 12 cycles of 20 s at 98 °C, 30 s at 60 °C and 8 min at 72 °C, with a final elongation at 72 °C for 10 min. PCR products were purified using AMPure XP beads (Beckman) for subsequent experiments. All related oligos are provided in Supplementary Table 1.

**gRNA design and Cas9-gRNA assembly**. The guide RNAs (gRNAs) were composed of CRISPR RNAs (crRNAs, GenScript) and trans-activating crRNAs (tracrRNAs, GenScript). The crRNAs were designed using the IDT online tool. Two of them targeted Alu elements in the consensus region of Alu subfamilies, and the other two targeted L1 elements at ORF0 and the 3'-ends of full-length L1Hs (Supplementary Table 1).

Equal molar amounts of tracrRNA and the corresponding crRNA were mixed together to a final concentration of 10 µM. After denaturation at 95 °C for 5 min, the gRNA duplex was formed during incubation at room temperature for at least 5 min. The Cas9-gRNA complex was assembled with 2 µM gRNA duplex and 1 µM HiFi Cas9 Nuclease V3 (Integrated DNA Technologies, 1081060) in CutSmart Buffer (NEB, B7204) at room temperature for 20 min. The Cas9-gRNA complex was stored at −20 °C for no more than two weeks before use.

**Target enrichment protocol adapted directly from nCATS (control)**. A total of 2 μg of amplified cDNA was used for target enrichment. After the dephosphorylation of the 5'-ends, DNA molecules were cleaved with 10 pmol of Cas9-gRNA mix (2.5 pmol each) at 37 °C for 15 min. Then, 1 μL of 10 mM dATP (NEB, N0440) and 5 U of Taq DNA polymerase (Vazyme, P101-d1-AC) were added to the sample for dA-tailing at 72 °C for 5 min. The product was used directly for adapter ligation by SQK-LSK110 (ONT) or barcoding by EXP-NBD114.

**Target enrichment protocol in capTEs**. To inactivate the 3'-hydroxyl terminus of DNA molecules, a single ddGMP was added to the ends of 2 μg of amplified cDNA in a 50 μL mixture containing 0.1 mM ddGTP (Roche, 03732738001), 0.25 mM CoCl$_2$ (NEB, M0315), 20 U of Terminal Transferase (NEB, M0315) and 1× TdT buffer (NEB, M0315). The reaction was performed at 37 °C for 2 h and stopped by heating for 20 min at 75 °C. Thereafter, the DNA sample was purified using AMPure XP beads and eluted in 24 μL of nuclease-free water. Three microliters of Quick CIP enzyme (NEB, M0525) and 3 μL of 10× Cutsmart buffer (NEB, B7204) were added to the eluate, and the mixture was incubated at 37 °C for 30 min for dephosphorylation. After inactivation of the phosphatase at 80 °C for 5 min, the DNA molecules were cut with 10 pmol of Cas9-gRNA complex (2.5 pmol each) at 37 °C for 15 min. For the removal of Cas9 protein from cleavage sites, the sample was digested with 1.5 μL of protease (Qiagen, 19155) for 10 min at 56 °C, followed by 70 °C for 15 min to inactivate protease. dA-tailing was carried out at 72 °C for 5 min with 1 μL of 10 mM dATP (NEB, N0440) and 5 U of Taq DNA polymerase (Vazyme, P101-d1-AC). The product was used directly for adapter ligation through SQK-LSK110 (ONT) or barcoding through NBD114 (ONT).

**Adapter ligation and sequencing library preparation for enriched samples**

*Library preparation with SQK-LSK110*. Sequencing adapters were ligated to DNA ends during a 10-min incubation at room temperature in an 80 μL reaction mixture containing dA-tailed DNA from target enrichment, 3.5 μL of AMX-F (ONT, SQK-LSK110), 10 μL of Quick T4 DNA Ligase (NEB, M2200) and 1× LNB (ONT, SQK-LSK110). The sample was then cleaned up using 0.4× AMPure XP beads with two washes with SFB (ONT, SQK-LSK110) before elution in 17 μL of EB (ONT, SQK-LSK110). The eluted DNA was ready for sequencing after mixing with 37.5 μL of SBII (ONT, SQK-LSK110) and 20.5 μL of LBII (ONT, SQK-LSK110).

*Multiplexed library preparation with EXP-NBD114*. One microgram of each enriched sample was barcoded using EXP-NBD114 in a 50 μL of reaction mixture at room temperature for 15 min. The components of the 50 μL reaction mixture were as follows: 1× Quick Ligation Reaction Buffer (NEB, M2200), 5 μL of Quick T4 DNA Ligase (NEB, M2200), 2.5 μL of barcode (ONT, EXP-NBD114) and 1.2 μg of enriched DNA. The sample was then cleaned up using 0.4× AMPure XP beads. A total of 2.5 μg of barcoded DNA was used for adapter ligation as follows: 5 μL of AMII (ONT, EXP-NBD114), 10 μL of Quick T4 DNA Ligase (NEB, M2200), and 1× LNB (ONT, SQK-LSK110) were added to barcoded DNA in a total volume of 100 μL, followed by a 10-min incubation at room temperature. The sample was cleaned using 0.4× AMPure XP beads and used for sequencing.

**Library preparation for total RNA-seq**. Amplified cDNAs were used for library preparation by SQK-LSK110 (ONT) following the manufacturer's instructions. Briefly, approximately 300 ng of cDNA was subjected to dA tailing in a 60 μL reaction mixture containing 3.5 μL of NEBNext FFPE DNA Repair Buffer (NEB, M6630), 2 μL of NEBNext FFPE DNA Repair Mix (NEB, M6630), 3.5 μL Ultra II End-prep reaction buffer (NEB, E7546) and 3 μL of Ultra II End-prep enzyme mix (NEB, E7546). The reaction was carried out at 20 °C for 5 min, followed by incubation at 65 °C for 5 min. The dA-tailed DNA sample was purified with 1× AMPure XP beads. Adapter ligation was performed as follows: In a total volume of 100 μL, the purified DNA sample was combined with 1× LNB (ONT, SQK-LSK110), 5 μL of AMX-F (ONT, SQK-LSK110), and 10 μL of Quick T4 DNA Ligase (NEB, M2200). The mixture was then incubated at room temperature for 10 min. After purification using 0.4× AMPure XP beads, the ligated sample are ready for sequencing.

**Incorporation of synthetic TE cDNAs**. Each synthesized TE cDNA consisted of an *E. coli* genomic sequence and a human Alu sequence with a gRNA target region. These cDNAs were diluted as shown in Supplementary Table 1 and mixed in equal volume. We incorporated 0.67 ng of the spike-in mix per 1 μg of full-length human cell cDNA.

**qPCR validation of gene expression**. The oligos used for qPCR are provided in Supplementary Table 1. For each selected gene, six biological replicates were performed using iTaq™ Universal SYBR® Green Supermix (Bio-Rad, 1725121). The thermal cycling protocol on a CFX96™ Real-Time System (Bio-Rad) was as follows: 3 min at 95 °C for initial denaturation, 41 cycles of 10 s at 95 °C, 30 s at 60 °C and 15 s at 72 °C, and 65 °C to 95 °C in 0.5 °C increments for the melting curve. The Cq value was used for quantitative analysis.

**Quantification of data yields**. The total number of available pores remained stable during the first hour of sequencing. We defined the data yield as the read output per available pore per minute for the first half hour. The relative data yield was calculated by normalizing to the control.

**Nanopore sequencing and data processing**. Nanopore sequencing was performed on a ONT GridION sequencer using MinION flow cells (R9.4.1). The electrical data were base-called and processed into FASTQ files using MinKNOW v.20.10.6 integrated with Guppy v4.2.3. Sequencing adapters and barcodes were trimmed from FASTQ format data using Porechop v0.2.4 (https://github.com/rrwick/Porechop) with the default parameters. The trimmed reads were filtered using NanoFilt[46] v2.8.0 with the commands --quality 7 --length 300. Next, reads were aligned to the human reference genome and the human reference transcriptome (hg38) using minimap2[47] v2.17 with the parameters "-ax splice" and "-ax map-ont" respectively. The resulting SAM file was converted to a BAM file and then sorted and indexed using SAMtools[48] v1.11.

**Analysis of strand bias**. Strand preferences were determined using published methods[19]. Reads were aligned to the consensus sequence of Alu and L1Hs. For Alu and L1Hs, only the first 30 and 80 base pairs of the 5' ends of each read were used for alignment, respectively. Local pairwise alignments were performed using the Bio.Align package from Biopython[49]. To determine sequence orientation, each read was aligned to the transposon consensus sequence as well as the reverse complement. To obtain high-confidence alignments, the match score, mismatch score, open gap score and extended gap score were set to 5, −5, −10 and −5, respectively. Finally, the alignment scores of Alu and L1Hs were at least 80 and 100, respectively. When

the alignment started at the first base of the read, it was accounted for.

**Identification of TEs in reads**. We scanned repeat elements in reads using RepeatMasker[50] v4.1.2. Here, interspersed repeats were annotated when the divergence from the consensus sequence was less than 18% and the Smith–Waterman score exceeded 225[51]. Reads containing identifiable Alu or L1 elements, as determined by RepeatMasker, were considered as TE-containing reads.

The age of TEs was determined based on the milliDivergence of TEs from the RepeatMasker annotation using the Jukes–Cantor model. Young TEs were defined as those with an age of less than two million years.

**Determination of relative data requirements for identifying transcripts containing target TEs**. We first performed saturation analysis using sufficient data. Specifically, we subsampled reads in different proportions (from 0.05 to 1 in intervals of 0.05) from the capTEs or total RNA-seq data pools of K562 samples. Each proportion was subsampled 50 times. The TE-containing transcripts were determined using reads containing target TEs. The data requirements for total RNA-seq relative to capTEs were based on the saturation curve equation.

**Analysis of noncanonical transcripts**. The mapping results of each cell line were integrated to assemble a refined transcriptome using StringTie[52] v2.1.7. For optimal outcomes, the StringTie procedure was repeated with a discontinuous threshold to determine the minimum coverage (from 1 to 20 at the interval of 1) required for the transcript assembly and the maximum locus gap (from 0 to 50 at the interval of 5) within which mapped reads were allowed to be merged. Thereafter, the minimum coverage was set to five and the maximum locus gap was set to twenty. GffCompare[53] (v0.12.6) was used to compare each of the assembled GTF files with the reference annotation GTF (Gencode annotation of the human transcriptome version 37). Based on the genomic context, we classified noncanonical transcripts into four categories, including genic (located within genic regions but not fully contained within introns), intronic (fully contained within introns), intergenic (fully contained within intergenic regions) and spanning intergenic and genic (across both intergenic and genic regions).

Alternative splicing and transcription initiation or termination are two major approaches for TEs to generate noncanonical transcripts. To investigate the roles of TEs in producing noncanonical transcripts, we reanalyzed and simplified the categories of noncanonical transcripts reported by GffCompare into four categories: intron retention (m and n, with at least one full-length intronic segment retained), noncanonical splicing (j, with at least one unannotated splicing junction), alternative TSS (i or u, fully contained in intronic or intergenic regions and started with TEs) and alternative TES (i or u, fully contained in intronic or intergenic regions and ended with TEs).

To identify noncanonical transcripts containing target TEs, we utilized BEDTools[54] intersect with the default parameters to find the intersection between the genomic features of the target TEs identified by RepeatMasker[50] in the human genome and the noncanonical transcripts. The transcripts with an overlap of one or more base pairs were considered noncanonical TE-containing transcripts. Likewise, for each TE locus, canonical transcripts with at least one base pair overlap with the analyzed TE were recorded and considered TE-containing transcripts. The genes to which these TE-containing transcripts belong were designated as TE-hosting genes for the corresponding TE sites.

To investigate the correlation between the production of noncanonical transcripts and the expression changes of TE-hosting genes, we counted the noncanonical transcripts under their compared reference genes as reported by GffCompare. The relative number of noncanonical transcripts for each gene, calculated as the difference in noncanonical transcript counts between cancer cells and matched normal cells, was used for further analysis.

**Transcription modes of target TEs**. Autonomously transcribed TEs were defined as those that exhibited at least one base pair overlap with the TSSs of noncanonical transcripts fully contained within intronic and intergenic regions. In the case of intron retention, TEs overlapped noncanonical transcripts downstream of the TSSs, and the transcription pattern of the overlapped transcript was classified as intron retention. For all other expressed TEs that overlapped with canonical or noncanonical transcripts but did not fall into the above two scenarios, they were classified as undergoing passive transcription. The corresponding overlapped transcripts were then recorded under the respective transcription modes of the analyzed TEs based on the descriptions provided.

To assess the transcriptional contribution of individual over-expressed TE loci under specific transcription modes to the expression levels of their host genes, we quantified the total expression levels of the transcripts recorded under each particular mode of the analyzed TE. We then calculated the proportion of these transcript expression levels relative to the overall expression levels of the TE-hosting gene. This proportion was used as the transcriptional contribution in the analyzed transcription modes for a specific TE site.

**Identification of TE insertions**. The multiple sorted alignment files of each cell line were merged to produce a single BAM file using SAMtools[48]. PALMER[19,39] was used to detect TEs that had not been annotated in the human reference genome version hg38 with the merged BAM files (--type ALU/LINE). The threshold of supporting reads was determined by elbow estimation. For Alu insertions, six was set as the minimum number of supporting reads. For L1 insertions, five and four were set as the minimum number of supporting reads in K562 and other cell lines, respectively. Insertions located within 200 bp were merged. The TE-inserted genes were determined based on the insertion sites of target TEs as outputted by PALMER.

To explore the potential correlation between TE insertions and the expression levels of their inserted genes, we quantified the number of TE insertions and calculated the difference in TE insertion counts between the cancer cells and their matched normal cell for each TE-inserted gene. The resulting difference value was used as the relative number of TE insertions for further investigation.

**Distribution analysis of TE insertions across gene bodies**. The distribution of TE insertions in gene regions (TSS, gene body and TES) was calculated as follows. For each gene, the 1 kb regions upstream of the TSS and downstream of the TES were split into ten nonoverlapping windows, while the gene body was split into 100 windows. Then the TE insertion counts were calculated for each window.

**Enrichment analysis of TE insertions in oncogenes**. The list of oncogenes was obtained from a previously published work[7] and used for the analyses described next. We first calculated the proportion of TE insertions in oncogenes among all insertions and the proportion of expressed TEs in oncogenes among all

expressed TEs. The enrichment fold value was the ratio of the calculated TE insertion proportion to the expressed TE proportion. The $P$ value was determined by Chi-squared test between inserted TEs and all expressed TEs, and a significant change was defined according to a BH (Benjamini–Hochberg)-adjusted $P < 0.05$.

**Quantification of TE and gene expression**. The coordinates of TEs were identified by RepeatMasker[50] v4.1.2. TE expression was quantified using FeatureCounts[55] v2.0.3 with the parameter "-f -O --minOverlap 20 -M --fraction -s 0 -L". Here, multi-mapping reads were counted, carrying a fractional count of $1/(x*y)$, where x is the total number of alignments reported for the same read and y is the total number of features overlapping with the read. FeatureCounts[55] was run as described above for the quantification of whole-transcriptome expression.

To evaluate the impact of noncanonical transcripts and canonical transcripts on the expression changes of TE-hosting genes, we measured their expression levels in cancer cells and matched normal cells. Next, we calculated the differences in expression levels between cancer cells and normal cells for both noncanonical and canonical transcripts. The contribution of each type of transcript was determined by taking the ratio of its difference value to the sum of the difference values of both types of transcripts.

**Analysis of differentially expressed genes and TEs**. Prior to differential expression analysis, a filter was applied to exclude genes or TEs with low expression by requiring a raw read count of at least three in three samples. Expression normalization and differential expression analysis were performed in R using the Bioconductor package DESeq2[56] v1.26.0. MDA-MB-231 cells were compared with MCF 10 A cells, while HCT 116 cells were compared with NCM460 cells. Within each cancer type, raw $P$ values resulting from differential expression analysis were adjusted by the BH approach to control the false discovery rate (FDR). Differentially expressed genes and TEs were selected on the basis of an absolute log2-transformed fold change of >1 and an adjusted $P$ value of <0.05.

**Data processing of NGS RNA-seq**. NGS RNA sequencing data of MDA-MB-231 and MCF10A cells[40,57] were downloaded from NCBI SRA (GEO accession: GSE75168) and aligned to the human reference genome version hg38 using STAR v2.7.10[58]. After alignment, we performed the locus-specific quantification of TEs using two different tools. First, we employed Telescope v1.0.3[17] with the parameter "--reassign_mode average" to quantify TE expression at specific genomic loci. Additionally, we utilized TEcount from TEtranscripts v2.2.1b[12] with the default parameter settings. To adapt the TEtranscripts tool for quantifying the expression levels of individual TEs, we made necessary modifications to the annotation file by assigning a unique name to each TE locus based on its genomic location information. Quantification of transcript abundance was performed using FeatureCounts[55].

**Statistics and reproducibility**. Statistical analyses were performed using R version 4.2.2. To assess whether the overexpressed TEs were enriched in specific TE subfamilies compared to all expressed TEs, the statistical analysis was performed using Fisher's exact test. Chi-squared test was performed to test whether autonomously expressed TE loci were enriched in specific TE subfamilies. To test whether TE insertions were enriched in oncogenes compared to all expressed TEs, Chi-squared test was

performed. The obtained $P$ values were adjusted by the BH approach.

In the study, data points represent biological replicates. We collected capTEs data from three independent biological replicates of HCT 116, NCM460, MDA-MB-231 and MCF-10A cells. The qPCR analyses were conducted using six biological replicates.

**Reporting summary**. Further information on research design is available in the Nature Portfolio Reporting Summary linked to this article.

## Data availability
The total RNA-seq data and capTEs data produced in this study were generated using the ONT GridION sequencer with MinION flow cells (R9.4.1) and have been deposited into the Gene Expression Omnibus (GEO) database under the accession number GSE205935. Source data for Figs. 1–5 are provided in Supplementary Data 8. All relevant data are available from the corresponding author on reasonable request.

## Code availability
The source code[59] is available at: https://github.com/KeyingLu/capTEs.

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

## Acknowledgements

We thank Siyuan Huang, Li Hu, Lanfang Yuan, Ziting Feng for their help in data processing. This work was supported by the following grants: 82173383 from Chinese National Natural Science Foundation (D.X.); ZYYC20006 from 1·3·5 project for disciplines of excellence, West China Hospital, Sichuan University (D.X.).

## Author contributions

D.X. and X.L. conceived the project. D.X. supervised this project. X.L. designed experimental pipeline. X.L. and X.C. performed the experiments. K.L., X.L. and X.C. analyzed and illustrated the data. K.T. provided help on analyzing on-target efficiency and assembling noncanonical transcript. X.L., K.L., X.C. and D.X. contributed to the writing of this manuscript.

## Competing interests

The authors declare no competing interests.
