## [Peer Review File · Communications Biology]

Reviewers' comments:

Reviewer #1 (Remarks to the Author):

Li et al. show capTEs, a new method to quantify TE expression using long reads. The authors show substantial improvements in TE quantification with respect to total RNA-Seq. Overall, the text could be improved in terms of order and phrasing. I suggest the authors to ask for feedback from an external collaborator, and a professional English proofreading expert.

Major remarks

First, at one point in the text they mention the measurement of gene expression with capTEs. For example, in lines 217-219, the authors mention that capTE can also measure gene expression. This needs to be further clarified, as they mention the use of gRNAs that Alu and L1 TEs.

In line with the previous point, how could capTEs be adapted to profile other type of TEs? There should be a more thorough discussion on the limitations and requirements (or future steps) that would allow capTEs to profile other type of TEs.

Lines 142-144: The authors indicate that there is no enrichment at the subfamily level. Figure 2 says "class" on the x-axis. I suggest them to rephrase this, because it looks like a result that is not in agreement with the others. Maybe they can also show the bars in terms of absolute numbers too.

Lines 275-291: The authors should describe in more detail the results obtained here. They mention correlations for Figure 5g, but technically this isn't shown in the panel. Also, in Figure 5g it seems that the 0 is not corresponding to the white color (scale is not symmetric).

The authors used TETranscripts (line 229), with the reasoning that is the most used tool for TE expression analysis in NGS data. However, in terms of benchmarking, it ranks last among all the tools. See the work of Schwarz et al. (doi: 10.1093/bib/bbab417, not an author). Moreover, it is unclear how they used it, because TETranscripts doesn't report expression at the locus level. There is almost no information about this on the methodology, and this needs to be further clarified, and an additional tool, such as Telescope (not an author either) should also be tested.

The authors should add a new figure indicating an overview of the entire workflow, as it would help a lot in understanding it.

Lines 471-472: What's the reasoning for imposing a divergence cutoff of 18% and alignment cutoff of 225? Do the authors have any evidence supporting the choice of these values?

Supplementary files should be self-contained (currently, they have no title and no description inside the file).

Minor remarks

Line 240: "The previous study" should be changed to "A previous study".

Lines 54,74,83: "nanopore" should be "Nanopore".

The labels of the panels should read left to right. For example, in Figure 4 and Figure 5, panels a, b and c go from top to bottom instead of left to right.

Lines 250-251: "autonomous degree of transcription" should be "degree of autonomous transcription". Though I would suggest them to rename it as "ratio of autonomous transcription"

Lines 268-269: What do the authors mean with "host genes"? For that result, do they only profile those TEs inside genes?

Line 471: "consensus seq" should be replaced to "consensus sequence".

Reviewer #2 (Remarks to the Author):

Transposable elements have gained increasing attentions in recent years. However, their expression analysis remains challenging, largely due to their repetitive nature. In this manuscript, Xie and colleagues present capTEs, which is a novel method combining Cas9-assisted TE enriched and nanopore long-read sequencing for locus-specific analysis of reference and non-reference TEs. They have carefully validated the efficiency and accuracy of capTEs for TE expression analysis, and further applied it to analyze the function of TEs in multiple human cancers.

Overall, I think this is a very interesting study, and I expect the presented method would be meaningful for the studying of transposable elements. While there are lots of computational tools designed for TE expression analysis in recent years, the improvement of related experimental techniques lag behind. Among the couple of nanopore-based protocols for TE expression analysis, the presented capTEs method has its advantages on data yield, cost and the power for analyzing young and non-reference TEs.

The manuscript is well-written, and the results are properly presented. I have a few comments for further improvement:

Major:

1. Line 229-232, it is confusing that Tetranscripts was applied for locus-specific TE analysis. As I know, Tetranscripts is designed for family-level analysis, while other tools such as SQuIRE or Telescope are for locus-specific TE analysis. More details can be found from the original paper as well as the comparative paper by Schwart et al. Brief Bioinform, 2022 (PMID: 34664075). The authors need to make sure that the correct tool was used, and observed change of differentially expressed TE loci was not an artifact.
2. Line 302-307 (also abstract, line 327, lines 563-564), the authors reported that the genomic locations of TE insertions are enriched surrounding the 3'UTR, which differ from the background (expressed TEs). While this is an interesting observation, the authors should be aware that: 1) the genomic distribution also differ among different classes (eg. LINEs, SINEs, ERVs) or even families of TEs; 2) it is possible that some specific TE families, particularly young TEs, prefer to form new insertions, while others prefer to get expressed. So is it possible the observed difference of genomic distribution is because due to the enrichment of specific TE classes/families, instead of they are expressed or newly inserted? The authors should take this point into account.

Minor:

Line 20: "functional output" is misleading. Do you mean "transcriptional output"?

Line 22: it seems "effects" is an overstatement. The "effects" have not been validated by experiments yet, and current data only support their correlation

Line 24: "in-depth" should be "high-depth"

Line 27: "in the active state" is confusing. Could the authors explain further?

Line 37: "promoters and cis-regulatory elements" are partly redundant. Promoters belong to cis-regulatory elements.

Line 39: "Measuring transcriptional outcomes resulting from TEs is thus a prerequisite" is inaccurate. There are tons of TEs serve as cis-elements, and I don't think transcriptional analysis is necessary for study such TEs.

Line 41: "transcript diversity" is confusing. Could the authors further explain?
Line 55: "structure variation" should be "structural variation"
Line 172: "most majority" should be "majority"
Line 495: "respectively" should be deleted
Line 515: "were" should be "that"
Line 562: hg38 or GRCh38? It should remain consistent across the manuscript.
Line 380: how the consensus sequences are obtained or generated?
Line 881: "Alu (light blue)", the color should be grey?
Line 917: "white" should be "grey"?

Referee comments

Reviewer #1 (Remarks to the Author):

Li et al. show capTEs, a new method to quantify TE expression using long reads. The authors show substantial improvements in TE quantification with respect to total RNA-Seq. Overall, the text could be improved in terms of order and phrasing. I suggest the authors to ask for feedback from an external collaborator, and a professional English proofreading expert.

Major remarks

1. First, at one point in the text they mention the measurement of gene expression with capTEs. For example, in lines 217-219, the authors mention that capTE can also measure gene expression. This needs to be further clarified, as they mention the use of gRNAs that Alu and L1 TEs.
2. In line with the previous point, how could capTEs be adapted to profile other type of TEs? There should be a more thorough discussion on the limitations and requirements (or future steps) that would allow capTEs to profile other type of TEs.
3. Lines 142-144: The authors indicate that there is no enrichment at the subfamily level. Figure 2 says “class” on the x-axis. I suggest them to rephrase this, because it looks like a result that is not in agreement with the others. Maybe they can also show the bars in terms of absolute numbers too.
4. Lines 275-291: The authors should describe in more detail the results obtained here. They mention correlations for Figure 5g, but technically this isn't shown in the panel. Also, in Figure 5g it seems that the 0 is not corresponding to the white color (scale is not symmetric).
5. The authors used Tetranscripts (line 229), with the reasoning that is the most used tool for TE expression analysis in NGS data. However, in terms of benchmarking, it ranks last among all the tools. See the work of Schwarz et al. (doi: 10.1093/bib/bbab417, not an author). Moreover, it is unclear how they used it, because Tetranscripts doesn't report expression at the locus level. There is almost no information about this on the methodology, and this needs to be further clarified, and an additional tool, such as Telescope (not an author either) should also be tested.
6. The authors should add a new figure indicating an overview of the entire workflow, as it would help a lot in understanding it.
7. Lines 471-472: What's the reasoning for imposing a divergence cutoff of 18% and alignment cutoff of 225? Do the authors have any evidence supporting the choice of these values?

8. Supplementary files should be self-contained (currently, they have no title and no description inside the file).

Minor remarks

1. Line 240: “The previous study” should be changed to “A previous study”.
2. Lines 54,74,83: “nanopore” should be “Nanopore”.
3. The labels of the panels should read left to right. For example, in Figure 4 and Figure 5, panels a, b and c go from top to bottom instead of left to right.
4. Lines 250-251: “autonomous degree of transcription” should be “degree of autonomous transcription”. Though I would suggest them to rename it as “ratio of autonomous transcription”
5. Lines 268-269: What do the authors mean with “host genes”? For that result, do they only profile those TEs inside genes?
6. Line 471: “consensus seq” should be replaced to “consensus sequence”.

Reviewer #2 (Remarks to the Author):

Transposable elements have gained increasing attentions in recent years. However, their expression analysis remains challenging, largely due to their repetitive nature. In this manuscript, Xie and colleges present capTEs, which is a novel method combining Cas9-assisted TE enriched and nanopore long-read sequencing for locus-specific analysis of reference and non-reference TEs. They have carefully validated the efficiency and accuracy of capTEs for TE expression analysis, and further applied it to analyze the function of TEs in multiple human cancers.

Overall, I think this is a very interesting study, and I expect the presented method would be meaningful for the studying of transposable elements. While there are lots of computational tools designed for TE expression analysis in recent years, the improvement of related experimental techniques lag behind. Among the couple of nanopore-based protocols for TE expression analysis, the presented capTEs method has its advantages on data yield, cost and the power for analyzing young and non-reference TEs.

The manuscript is well-written, and the results are properly presented. I have a few comments for further improvement:

Major:

1. Line 229-232, it is confusing that Tetranscripts was applied for locus-specific TE analysis. As I know, Tetranscripts is designed for family-level analysis, while other tools such as SQuIRE or Telescope are for locus-specific TE analysis. More details can be found from the original paper as well as the comparative paper by Schwart et al. Brief Bioinform, 2022 (PMID: 34664075). The authors need to make sure that the correct tool was used, and observed change of differentially expressed TE loci was not an artifact.

2. Line 302-307 (also abstract, line 327, lines 563-564), the authors reported that the genomic locations of TE insertions are enriched surrounding the 3'UTR, which differ from the background (expressed TEs). While this is an interesting observation, the authors should be aware that: 1) the genomic distribution also differ among different classes (eg. LINEs, SINEs, ERVs) or even families of TEs; 2) it is possible that some specific TE families, particularly young TEs, prefer to form new insertions, while others prefer to get expressed. So is it possible the observed difference of genomic distribution is because due to the enrichment of specific TE classes/families, instead of they are expressed or newly inserted? The authors should take this point into account.

Minor:

1. Line 20: “functional output” is misleading. Do you mean “transcriptional output”?

2. Line 22: it seems “effects” is an overstatement. The “effects” have not been validated by experiments yet, and current data only support their correlation

3. Line 24: “in-depth” should be “high-depth”

4. Line 27: “in the active state” is confusing. Could the authors explain further?

5. Line 37: “promoters and cis-regulatory elements” are partly redundant. Promoters belong to cis-regulatory elements.

6. Line 39: “Measuring transcriptional outcomes resulting from TEs is thus a prerequisite” is inaccurate. There are tons of TEs serve as cis-elements, and I don't think transcriptional analysis is necessary for study such TEs.

7. Line 41: “transcript diversity” is confusing. Could the authors further explain?

8. Line 55: “structure variation” should be “structural variation”

9. Line 172: “most majority” should be “majority”

10. Line 495: “respectively” should be deleted

11. Line 515: “were” should be “that”
12. Line 562: hg38 or GRCh38? It should remain consistent across the manuscript.
13. Line 380: how the consensus sequences are obtained or generated?
14. Line 881: “Alu (light blue)”, the color should be grey?
15. Line 917: “white” should be “grey”?

Reviewer #1 (Remarks to the Author):

Li et al. show capTEs, a new method to quantify TE expression using long reads. The authors show substantial improvements in TE quantification with respect to total RNA-Seq. Overall, the text could be improved in terms of order and phrasing. I suggest the authors to ask for feedback from an external collaborator, and a professional English proofreading expert.

RESPONSE: We thank the reviewer for providing constructive feedback. We have had the manuscript professionally edited by American Journal Experts (AJE) to improve the language and grammar, ensuring clear communication of our research.

Below are our point-by-point responses. Modified text is in **blue**.

Major remarks

1. First, at one point in the text they mention the measurement of gene expression with capTEs. For example, in lines 217-219, the authors mention that capTE can also measure gene expression. This needs to be further clarified, as they mention the use of gRNAs that Alu and L1 TEs.

RESPONSE: We agree and have revised our description as follows:

Page 12 (lines 221-223): To test the reliability of the quantitative results, we first validated the expression levels of genes containing Alu or L1 targeted by our designed gRNAs.

2. In line with the previous point, how could capTEs be adapted to profile other type of TEs? There should be a more thorough discussion on the limitations and requirements (or future steps) that would allow capTEs to profile other type of TEs.

RESPONSE: We thank the reviewer for the comments. We have added this content to the Discussion.

Page 19 (lines 381-396) Discussion: The capTEs method has been validated based on two distinct TE families, Alu and L1, which have very different genomic distributions. However, to apply this method to other types of TEs, we suggest the application of several tests to ensure proper execution and feasibility, including tests of gRNA cutting efficiency, the positional distribution of targets in reads, and strand bias. In theory, if targeted enrichment is successful, the target sequence should be concentrated at the beginning and ends of the reads. Although our method considerably improves strand bias, we found that the nucleotide sequence at the cutting site of Cas9-gRNA complexes may affect the strand distribution to some extent, as observed in our preliminary testing of different gRNAs. Furthermore, if quantitative analysis of TE expression is desired, we recommend incorporating a standard containing known concentrations of the target sequence for quality control. Currently, our method produces data with an approximately 80% unique mapping rate, comparable to total RNA-seq data produced on the Nanopore sequencing platform. However, accurately quantifying TE expression levels requires further improvement, which is expected to be achieved by introducing UMIs and enhancing sequencing accuracy.

3. Lines 142-144: The authors indicate that there is no enrichment at the subfamily level. Figure 2 says “class” on the x-axis. I suggest them to rephrase this, because it looks like a result that is not in agreement with the others. Maybe they can also show the bars in terms of absolute numbers too.

RESPONSE: We have replaced “class” with “subfamilies” and labeled the proportion of each TE subfamily in Fig. 2. We have reworded the relevant text accordingly.

Page 36 Fig. 2d:

d, Bar plots showing the subfamily proportions of target TEs detected by capTEs (orange) and total RNA-seq (green).

Page 8 (lines 146-148): We next investigated whether capTEs exhibited a bias toward some

specific TE subfamilies within the target TEs.

4. Lines 275-291: The authors should describe in more detail the results obtained here. They mention correlations for Figure 5g, but technically this isn't shown in the panel. Also, in Figure 5g it seems that the 0 is not corresponding to the white color (scale is not symmetric).

RESPONSE: We appreciate the comments, and we have expanded the description of the results related to the transcription modes of TEs. Regarding Figure 5g, we have added the correlation coefficient to the figure and modified the legend to ensure that 0 corresponds to white, thus improving the clarity and accuracy of the representation.

Page 15-16 (lines 294-320) Characterizing the impacts of internal and inserted TEs on the transcriptome of cancer cells: To determine whether differentially expressed TEs are involved in generating noncanonical transcripts, we analyzed the expression levels of each TE locus and the corresponding TE-derived transcripts that began or ended with the TE of interest. As expected, the density of TE-derived transcripts changed along with the expression levels at the TE locus, with higher expression levels corresponding to higher transcript densities (Fig. 5d). These results indicate a role of TEs in regulating gene expression.

To investigate how individual TEs contribute to changes in gene expression, we divided the transcription of each overexpressed TE into three modes based on the transcription patterns of the corresponding TE transcripts: autonomous transcription, intron retention, and passive transcription. We then calculated the proportion of transcript abundance for each transcription mode, which we used as the transcriptional contribution of that particular mode. Our analysis revealed that, consistent with a previous study, the majority of TE RNAs originated from the promoter activity of host genes. We observed that upregulated TE loci were predominantly transcribed through intron retention (14%) and passive transcription (65%) (Fig. 5e). Furthermore, we found that 19% of TEs were transcribed via more than one transcription mode (Fig. 5e). Interestingly, we identified over 200 overexpressed TE loci where transcription was initiated independently, and some of them were located within the gene bodies of cancer-related genes. For example, AluY (chr19: 4099969-4100275) and L1PA4 (chr14: 61707309-61708779) initiated transcription in the genomic regions of *MAP2K2* (Mitogen-activated protein kinase 2) and *HIF1A* (hypoxia-inducible factor 1 subunit alpha), respectively, and generated noncanonical transcripts (Fig. 5f). These transcripts accounted for approximately 49% and 50% of the total expression levels of their host genes, *MAP2K2* and *HIF1A*, respectively (Fig. 5e). However, the TE-initiated transcripts had nonidentical sequences compared to the classical transcripts (Fig. 5f); thus, their functions require further investigation.

Page 40 Fig. 5g:

g, Heatmap showing the relative frequency of cancer cell-specific insertions in cancer cells compared to matched normal cells (left) and transcriptional changes in host genes (right). The color bar displays insertions in oncogenes (green). r_s (Spearman's correlation coefficient) between the relative frequency of TE insertions and changes in gene expression are 0.69 for MDA-MB-231 cells and 0.62 for HCT 116 cells.

5. The authors used Tetrascripts (line 229), with the reasoning that is the most used tool for TE expression analysis in NGS data. However, in terms of benchmarking, it ranks last among all the tools. See the work of Schwarz et al. (doi: 10.1093/bib/bbab417, not an author). Moreover, it is unclear how they used it, because Tetrascripts doesn't report expression at the locus level. There is almost no information about this on the methodology, and this needs to be further clarified, and an additional tool, such as Telescope (not an author either) should also be tested.

RESPONSE: We thank the reviewer for this valuable suggestion. We have added information on the usage of Tetrascripts in the Methods. This tool can be adapted for the locus-specific analysis of TEs by assigning a unique symbol to each TE locus instead of using TE family names.

We have taken the reviewer's suggestion and conducted TE expression analysis using the suggested tool, Telescope. The results demonstrated that a greater number of TE loci were quantified by using our method than by NGS, especially young TEs. These findings align with our previous results obtained from Tetrascripts. We have updated the relevant parts of the manuscript to reflect this change (Fig 4c-e, Supplementary Figure 6, Supplementary Figure 7).

Page 29 (lines 625-632) Methods: After alignment, we performed the locus-specific quantification of TEs using two different tools. First, we employed Telescope v1.0.3 with the

parameter "--reassign_mode average" to quantify TE expression at specific genomic loci. Additionally, we utilized TEcount from Tetrascripts v2.2.1b with the default parameter settings. To adapt the Tetrascripts tool for quantifying the expression levels of individual TEs, we made necessary modifications to the annotation file by assigning a unique name to each TE locus based on its genomic location information.

Page 12-13 (lines 240-256) Locus-specific quantification of TE expression: We next compared our TE expression results to those obtained from NGS data using two independent TE-dedicated bioinformatics tools, Tetrascripts and Telescope. Our analysis revealed that capTEs detected expression changes at 65,631 TE loci between MDA-MB-231 and MCF 10A cells, which was 2.6 and 3.4 times higher than the number of TE loci reported by Tetrascripts and Telescope, respectively based on NGS data (Supplementary figure 6a, Fig. 4c). Importantly, capTEs exhibited a remarkable ability to detect young TEs, with the results exceeding the number of NGS-quantified loci by an average of twenty times (15-fold more than Tetrascripts and 25-fold more than Telescope, Supplementary figure 6b, Fig. 4d). We further compared our results to those reported by Telescope, as this tool was specifically designed for the locus-specific analysis of TE expression. Two pieces of evidence suggested that capTEs presented a higher sensitivity than NGS in detecting modest changes in the expression of individual TEs. First, capTEs identified more differential TEs at various evolutionary ages than NGS (Fig. 4e, Supplementary Figure 7a). Second, we observed lower expression changes for differential TEs detected only by capTEs in comparison to those revealed by both capTEs and NGS (Supplementary Figure 7b).

Page 38 Fig 4c-e:

c, Bar plot showing the number of TEs quantified with capTEs (blue) and NGS (orange). **d**, Bar plot showing the number of young TEs quantified with capTEs (blue) and NGS (orange). **e**, Line chart showing the number of differential TEs at various evolutionary ages identified with capTEs and NGS. Myr: million years. **c-e**, NGS data were analyzed using Telescope.

Page 7 Supplementary Materials, Supplementary Figure 6:

a, Bar plot showing the number of TEs quantified with capTEs (blue) and NGS (orange). **b**, Bar plot showing the number of young TEs quantified with capTEs (blue) and NGS (orange). **a-b**, NGS data were analyzed using TEtranscripts.

Page 8 Supplementary Materials, Supplementary Figure 7:

a, Bar plot showing the number of differential TE loci measured by capTEs and NGS. **b**, Box plot showing the expression fold changes of differential TEs codetected by capTEs and NGS (blue) and differential TEs detected only by capTEs (orange). The y-axis represents absolute values of log₂ expression fold change determined using capTEs data. The statistical analysis was performed using Wilcox rank sum test. **a-b**, NGS data were analyzed using Telescope.

6. The authors should add a new figure indicating an overview of the entire workflow, as it would help a lot in understanding it.

RESPONSE: We appreciate the helpful comment from reviewer #1, which has allowed us to better illustrate our methodology, including the experimental workflow and corresponding bioinformatics pipeline. In the previous version of our manuscript, we presented the experimental workflow in Figure 1a. In this revised version, we have added the workflow for data analysis, as shown in Supplementary Figure 1d.

Page 2 Supplementary Materials, Supplementary figure 1d:

d, Workflow for data analysis. The mapping results were obtained from raw signal data using the following steps: base-calling and FASTQ file generation with Guppy, trimming and filtering with Porechop and NanoFilt, alignment to the reference genome and transcriptome (hg38) using minimap2, and TE annotation using RepeatMasker. These results were then utilized for subsequent analysis, including quantifying the expression of TEs and genes, identifying noncanonical transcripts and detecting nonreference TE insertions.

7. Lines 471-472: What’s the reasoning for imposing a divergence cutoff of 18% and alignment cutoff of 225? Do the authors have any evidence supporting the choice of these values?

RESPONSE: We thank the reviewer for the question. The choice of a divergence cutoff of 18% and an alignment cutoff of 225 was based on the guidance provided on the RepeatMasker website (<https://www.repeatmasker.org/webrepeatmaskerhelp.html>). Specifically, the recommended divergence cutoff for most primate-specific repeats is 18%, which is the reason we chose this value for our analysis of TEs in human cells. The alignment cutoff of 225 was employed for two reasons. Firstly, it is the default cutoff score set by RepeatMasker. Secondly, it is applicable to most repeats, as the majority of false matches tend to have scores close to 225 and lower scores give more false matches.

We have added a reference to the RepeatMasker paper that provides details on the default alignment cutoff score (PMID: 19274634). We hope this clarifies our rationale for selecting these values.

8. Supplementary files should be self-contained (currently, they have no title and no description inside the file).

RESPONSE: We have modified the Supplementary files accordingly.

Minor remarks

1. Line 240: “The previous study” should be changed to “A previous study”.

RESPONSE: We have modified the text accordingly.

Page 13 (lines 257-258): A previous study reported the overexpression of young TEs in cancers, possibly due to the loss of DNA methylation in surrounding genomic regions.

2. Lines 54,74,83: “nanopore” should be “Nanopore”.

RESPONSE: We have modified the text accordingly.

Page 4 (lines 53-54) Introduction: Nanopore sequencing combined with CRISPR/Cas9 technology has been used to study different types of genomic variants, such as structural variations.

Page 5 (lines 75-77): We developed capTEs as a strategy for enriching and identifying TE transcripts by selectively detecting the flanking regions of known TEs using Nanopore sequencing (Fig 1a).

Page 5 (lines 85-86): After sequencing on the ONT platform, we assembled these long reads into transcripts for subsequent analysis.

3. The labels of the panels should read left to right. For example, in Figure 4 and Figure 5, panels a, b and c go from top to bottom instead of left to right.

RESPONSE: We have reorganized the panels to ensure a labeling order from left to right as much as possible.

4. Lines 250-251: “autonomous degree of transcription” should be “degree of autonomous transcription”. Though I would suggest them to rename it as “ratio of autonomous transcription”

RESPONSE: We agree with this suggestion and have modified the text accordingly.

Page 14 (lines 267-270): We further characterized the ratio of autonomous transcription at each locus in driving transcription (i.e., the autonomous transcription level of the given locus relative to the additive abundance of all transcripts containing that TE) (Fig. 4h).

5. Lines 268-269: What do the authors mean with “host genes”? For that result, do they only profile those TEs inside genes?

RESPONSE: We thank the reviewer for this question. We defined the host gene of a specific TE as a gene whose transcript overlapped with that of the TE in the genome. In this study, we measured the expression levels of all targeted TEs (Alu and L1), some of which were located in intergenic regions. To investigate the relationship between the expression of TEs and genes,

we focused on the TEs that overlapped with genes.

Page 29 (lines 609-610) Methods: The host gene of a specific TE was defined as a gene whose transcript overlapped with that of the TE in the genome.

6. Line 471: “consensus seq” should be replaced to “consensus sequence”.

RESPONSE: We have modified the text accordingly.

Page 25 (lines 526-528) Methods: Here, interspersed repeats were annotated when the divergence from the consensus sequence was less than 18% and the Smith–Waterman score exceeded 225.

Reviewer #2 (Remarks to the Author):

Transposable elements have gained increasing attentions in recent years. However, their expression analysis remains challenging, largely due to their repetitive nature. In this manuscript, Xie and colleges present capTEs, which is a novel method combining Cas9-assisted TE enriched and nanopore long-read sequencing for locus-specific analysis of reference and non-reference TEs. They have carefully validated the efficiency and accuracy of capTEs for TE expression analysis, and further applied it to analyze the function of TEs in multiple human cancers.

Overall, I think this is a very interesting study, and I expect the presented method would be meaningful for the studying of transposable elements. While there are lots of computational tools designed for TE expression analysis in recent years, the improvement of related experimental techniques lag behind. Among the couple of nanopore-based protocols for TE expression analysis, the presented capTEs method has its advantages on data yield, cost and the power for analyzing young and non-reference TEs.

The manuscript is well-written, and the results are properly presented. I have a few comments for further improvement:

RESPONSE: We thank the reviewer for the positive assessment of our work and valuable suggestions for improving our manuscript. We have carefully considered each comment. Below are our point-by-point responses. Modified text is in blue.

Major:

1. Line 229-232, it is confusing that Tetranscripts was applied for locus-specific TE analysis. As I know, Tetranscripts is designed for family-level analysis, while other tools such as SQuIRE or Telescope are for locus-specific TE analysis. More details can be found from the original paper as well as the comparative paper by Schwart et al. Brief Bioinform, 2022 (PMID: 34664075). The authors need to make sure that the correct tool was used, and observed change

of differentially expressed TE loci was not an artifact.

RESPONSE: We appreciate the constructive feedback. As suggested by the reviewer, we performed an additional TE expression analysis using Telescope, a specialized tool for locus-specific TE analysis. The reanalysis confirmed our previous findings and showed that more TE loci were quantified by using our method than by NGS, particularly young TEs. We have updated the relevant parts of the manuscript to reflect this change (Fig. 4c-e, Supplementary Figure 6,7).

Furthermore, to ensure the reliability of our quantitative results, we generated heatmap displaying the expression levels of differentially expressed TEs in the tested cells (shown below) and observed consistent expression changes of these differentially expressed TEs across three biological replicates.

a, Heatmap displaying the expression levels of differential TEs in MDA-MB-231 and MCF 10A cells. The quantitative results were obtained by Telescope using NGS data. **b**, Heatmap displaying the expression levels of differential TEs in MDA-MB-231 and MCF 10A cells using capTEs data.

Page 12-13 (lines 240-256) Locus-specific quantification of TE expression: We next compared our TE expression results to those obtained from NGS data using two independent TE-dedicated bioinformatics tools, Tetranscripts and Telescope. Our analysis revealed that capTEs detected expression changes at 65,631 TE loci between MDA-MB-231 and MCF 10A cells, which was 2.6 and 3.4 times higher than the number of TE loci reported by Tetranscripts and Telescope, respectively based on NGS data (Supplementary figure 6a, Fig. 4c). Importantly, capTEs exhibited a remarkable ability to detect young TEs, with the results exceeding the number of NGS-quantified loci by an average of twenty times (15-fold more than Tetranscripts and 25-fold more than Telescope, Supplementary figure 6b, Fig. 4d). We further compared our results to those reported by Telescope, as this tool was specifically designed for the locus-specific analysis of TE expression. Two pieces of evidence suggested that capTEs presented a higher sensitivity than NGS in detecting modest changes in the expression of individual TEs. First, capTEs identified more differential TEs at various evolutionary ages than NGS (Fig. 4e, Supplementary Figure 7a). Second, we observed lower expression changes for differential TEs

detected only by capTEs in comparison to those revealed by both capTEs and NGS (Supplementary Figure 7b).

Page 38 Fig 4c-e:

c, Bar plot showing the number of TEs quantified with capTEs (blue) and NGS (orange). **d**, Bar plot showing the number of young TEs quantified with capTEs (blue) and NGS (orange). **e**, Line chart showing the number of differential TEs at various evolutionary ages identified with capTEs and NGS. Myr: million years. **c-e**, NGS data were analyzed using Telescope.

Page 7 Supplementary Materials, Supplementary Figure 6:

a, Bar plot showing the number of TEs quantified with capTEs (blue) and NGS (orange). **b**, Bar plot showing the number of young TEs quantified with capTEs (blue) and NGS (orange). **a-b**, NGS data were analyzed using Tetrascripts.

Page 8 Supplementary Materials, Supplementary Figure 7:

a, Bar plot showing the number of differential TE loci measured by capTEs and NGS. **b**, Box plot showing the expression fold changes of differential TEs codetected by capTEs and NGS (blue) and differential TEs detected only by capTEs (orange). The y-axis represents absolute values of log₂ expression fold change determined using capTEs data. The statistical analysis was performed using Wilcox rank sum test. **a-b**, NGS data were analyzed using Telescope.

2. Line 302-307 (also abstract, line 327, lines 563-564), the authors reported that the genomic locations of TE insertions are enriched surrounding the 3'UTR, which differ from the background (expressed TEs). While this is an interesting observation, the authors should be aware that: 1) the genomic distribution also differ among different classes (eg. LINES, SINEs, ERVs) or even families of TEs; 2) it is possible that some specific TE families, particularly young TEs, prefer to form new insertions, while others prefer to get expressed. So is it possible the observed difference of genomic distribution is because due to the enrichment of specific TE classes/families, instead of they are expressed or newly inserted? The authors should take this point into account.

RESPONSE: We thank the reviewer for the constructive comments. We have conducted further analysis to address the concerns that were raised. Specifically, we analyzed the genomic text of Alu and L1 separately and found that the genomic locations of Alu insertions displayed a significant preference for 3'UTRs (Fig. 5h). To determine whether this distribution bias was specific to certain Alu subfamilies rather than insertion events, we further analyzed the genomic context of expressed Alu subfamilies, including Alu Y, Alu J, and Alu S. However, we did not detect any significant differences in the genomic distribution among these subfamilies when compared to the Alu family as a whole (Fig. 5i). Moreover, we consistently observed that inserted Alu elements exhibited a distinct preference for 3'UTRs compared to each of the examined Alu subfamilies (Supplementary Figure 8d). These results indicate that enrichment in the 3'UTR is a specific feature of expressed inserted Alu elements. We have included the additional results in the revised manuscripts.

Page 17 (lines 330-344) Characterizing the impacts of internal and inserted TEs on the transcriptome of cancer cells: To gain further insight, we analyzed the genomic context of

inserted and internal TEs by families, revealing the difference between nonreference and reference Alu. The density of inserted Alu elements peaked in TESs, while that of expressed internal Alu elements plateaued across the gene body (Fig. 5h). Furthermore, we observed that the genomic locations of Alu insertions displayed a significant preference for 3'UTRs, which are known to be enriched in expression quantitative trait loci (eQTL), compared to the background (expressed Alu) (Supplementary Figure 8d, Fig. 5i). To determine whether this distribution bias was due to specific Alu subfamilies instead of insertion, we further analyzed the genomic context of expressed Alu subfamilies, including Alu Y, Alu J, and Alu S. However, we did not detect significant differences in the proportion of 3'UTR between these subfamilies and the Alu family as a whole, and we repeatedly observed a distribution preference of inserted Alu for the 3'UTR compared to each of the Alu subfamilies described above (Figure 5i, Supplementary Figure 8d), indicating that enrichment in the 3'UTR is a specific feature of expressed inserted Alu.

Page 40 Fig. 5h-i:

h, Distribution of expressed inserted Alu (red) and expressed reference TEs (blue) in gene bodies. **i**, Stacked bar plots showing the proportion of genomic features (3'UTR, 5'UTR, intron, CDS and intergenic regions) in expressed inserted Alu and expressed reference Alu, Alu Y, Alu S and Alu J.

Page 9 Supplementary Materials, Supplementary Figure 8d:

d, Location enrichment of Alu insertions in 3'UTRs compared to reference annotated Alu as a whole or Alu subfamilies, including Alu Y, Alu S and Alu J. All Alu insertions identified in breast and colorectal cancer cells are included and the reference Alu refers to those successfully transcribed. Significant * represents BH-adjusted $P < 0.05$ reported by the Chi-squared test between Alu insertions and all expressed loci of Alu elements or subfamilies.

Minor:

1. Line 20: “functional output” is misleading. Do you mean “transcriptional output”?

RESPONSE: Yes. We have revised the text accordingly.

Page 2 (lines 18-23) Abstract: Here, we introduce a long-read targeted RNA sequencing method, Cas9-assisted profiling TE expression sequencing (capTEs), for quantitative analysis of transcriptional outputs for individual TEs, including transcribed nonreference insertions, noncanonical transcripts from various transcription patterns and their correlations with expression changes in related genes.

2. Line 22: it seems “effects” is an overstatement. The “effects” have not been validated by experiments yet, and current data only support their correlation

RESPONSE: We have modified the text to “their correlations with expression changes in related genes”

Page 2 (lines 18-23) Abstract: Here, we introduce a long-read targeted RNA sequencing method, Cas9-assisted profiling TE expression sequencing (capTEs), for quantitative analysis of transcriptional outputs for individual TEs, including transcribed nonreference insertions, noncanonical transcripts from various transcription patterns and their correlations with expression changes in related genes.

3. Line 24: “in-depth” should be “high-depth”

RESPONSE: We have reworded this text.

Page 2 (lines 23-25) Abstract: This method selectively identified TE-containing transcripts and outputted data with up to 90% TE reads, maintaining a comparable data yield to whole-transcriptome sequencing.

4. Line 27: “in the active state” is confusing. Could the authors explain further?

RESPONSE: "In the active state" refers to the state in which TEs are involved in regulating gene expression. We have modified the text.

Page 1 (lines 25-27) Abstract: We applied capTEs to human cancer cells and found that internal and inserted Alu elements may employ distinct regulatory mechanisms to upregulate gene expression.

5. Line 37: “promoters and cis-regulatory elements” are partly redundant. Promoters belong to cis-regulatory elements.

RESPONSE: We have modified this text accordingly.

Page 3 (lines 35-37) Introduction: Furthermore, as TEs are a prolific source of cis-regulatory

elements, their insertion can introduce regulatory sequences, potentially affecting the expression of inserted genes.

6. Line 39: “Measuring transcriptional outcomes resulting from TEs is thus a prerequisite” is inaccurate. There are tons of TEs serve as cis-elements, and I don’t think transcriptional analysis is necessary for study such TEs.

RESPONSE: We have modified this sentence to “Measuring transcriptional outcomes resulting from TEs helps to broaden our understanding of how TEs regulate various biological processes”.

Page 3 (lines 37-39) Introduction: Therefore, measuring transcriptional outcomes resulting from TEs helps to broaden our understanding of how TEs regulate various biological processes.

7. Line 41: “transcript diversity” is confusing. Could the authors further explain?

RESPONSE: Herein, "transcript diversity" refers to the fact that TEs can be found in various types of transcripts, including canonical full-length TE transcripts, as well as chimeric transcripts originating from TE promoter activity or from passive cotranscription. This diversity in TE transcript structures and origins makes their analysis and quantification challenging. We have provided further clarification in the revised manuscript.

Page 3 (lines 40-42) Introduction: However, the repetitive nature of TEs and the presence of diverse types of TE transcripts complicate the detection and quantification of TEs at the transcriptional level.

8. Line 55: “structure variation” should be “structural variation”

RESPONSE: We have modified this text accordingly.

Page 4 (lines 53-54) Introduction: Nanopore sequencing combined with CRISPR/Cas9 technology has been used to study different types of genomic variants, such as structural variations.

9. Line 172: “most majority” should be “majority”

RESPONSE: We have modified this text accordingly.

Page 10 (lines 179-180) Introduction Identification of noncanonical transcripts and transcribed nonreference TEs: The majority of the identified noncanonical transcripts overlapped with genic regions.

10. Line 495: “respectively” should be deleted

RESPONSE: We have modified this text accordingly.

Page 26 (lines 550-551) Methods: The mapping results of each cell line were integrated to assemble a refined transcriptome using StringTie v2.1.7.

11. Line 515: “were” should be “that”

RESPONSE: We have modified this text accordingly.

Page 27 (lines 572-574) Methods: We defined autonomously transcribed TEs as those TEs located at the TSSs of the noncanonical transcripts.

12. Line 562: hg38 or GRCh38? It should remain consistent across the manuscript.

RESPONSE: We have standardized this as hg38.

Page 25 (lines 521-523) Methods: Next, reads were aligned to the reference genome and the reference transcriptome (hg38) using minimap2 v2.17 with the parameters "-ax splice" and "-ax map-ont" respectively.

Page 27 (lines 577-579) Methods: PALMER was used to detect TEs that had not been annotated in genome reference hg38 with the merged BAM files (--type ALU/LINE).

13. Line 380: how the consensus sequences are obtained or generated?

RESPONSE: We aligned the sequences of Alu subfamilies and observed that they share almost identical sequences of the first 60 nucleotides. Therefore, we designed sgRNAs targeting this region, as depicted in Supplementary Figure 1a.

14. Line 881: “Alu (light blue)”, the color should be grey?

RESPONSE: We have made the necessary correction and changed the color of Alu to gray.

Page 5 Supplementary Materials, Supplementary Figure 4a:

a, Pie charts showing the proportion of novel transcripts containing only Alu (gray), only L1 (yellow) and both Alu and L1 (orange) identified by capTEs and total RNA-seq, respectively.

15. Line 917: “white” should be “grey”?

RESPONSE: We have made the necessary correction and changed the “white” into “gray”.

Page 9 Supplementary Materials, Supplementary Figure 8: b, Stacked bar plot showing the number of TE insertions in genic (gray) and intergenic (orange) regions in cancer cells HCT 116 and MDA-MB-231.

Reviewers' comments:

Reviewer #1 (Remarks to the Author):

In the revised version of their work, Li et al. improved upon several of the points raised in the first review round. Although most of their answers are satisfactory, there are still some points that need to be improved for recommendation of the manuscript for publication in the journal.

Major remarks

I previously pointed out "In line with the previous point, how could capTEs be adapted to profile other type of TEs? There should be a more thorough discussion on the limitations and requirements (or future steps) that would allow capTEs to profile other type of TEs"

1. The authors discussed considerably the point I made before of using capTEs to detect other type of TEs. The discussion of this issue is an excellent addition to their work.

Later they mention that "Taking all detected TEs into account, we found no significant difference in the proportion of TE subfamilies between the TE loci identified by capTEs and those detected through total RNA-seq (Fig. 2d)". What is the total number of TEs detected with total RNA-Seq? What fraction of those are detected by capTEs? Since capTEs was used on specific subfamilies, it would be useful to know how many TEs would be missed when compared to a regular RNA-Seq experiment.

2. In the analysis of "Locus-specific quantification of TE expression ", the authors expanded their comparison of the TE expression obtained with capTEs to that obtained with Tetranscripts and Telescope, with the latter being suggested by me and the other reviewer. This provided another layer for their analysis, in which they also expanded upon.

Similar to the point raised before, how do the proportions of total detected TEs vary compared to those detected by Tetranscripts and by Telescope? Of those detected with the tools, what is the proportion of young vs old TEs?

Addressing and discussing this point would further highlight potential advantages of capTEs over specific subfamilies of young TEs.

3. In "Characterizing the impacts of internal and inserted TEs on the transcriptome of cancer cells", there are a few points that I don't see described in detail. How did the authors assess the overlaps between TEs and genes? What tool did they use? From what can be seen in the methods, this is not detailed.

In regards to this point, they also mention "We further analyzed TE transcripts that contained exonic Alu or L1 in Gencode annotations". How was this assessed?

When they mention "host genes" (lines 609-610), what tool was used? Were TEs fully contained within exons consider? In the case of exon-overlapping TEs (as those described above), was there any threshold considered regarding the amount of overlap between the TE an exons?

Minor remarks

Line 578: "genome reference hg38" to "Human reference genome version hg38"

Line 600-601: "The oncogenes referred to the published list". This should be moved to the beginning of the paragraph, and changed to "The list of oncogenes was obtained from a previously published

work (reference) and used for the analyses described next”.

Figure 4g. The color of the box plots in the inset depicting expression are not described in the legend

Figure 5g. Although the author implemented some of the suggestions made in the first round of revisions, now it is unclear how they fit both “Relative No. of TE insertions” and “gene expression changes (Log2 fold change) in the same scale of the plot. This should be clarified in detail in the legend and methods.

Reviewer #2 (Remarks to the Author):

I am satisfied with the authors' efforts for the revision. All my questions have been properly addressed, and I have no further comments.

Referee comments

Reviewer #1 (Remarks to the Author):

In the revised version of their work, Li et al. improved upon several of the points raised in the first review round. Although most of their answers are satisfactory, there are still some points that need to be improved for recommendation of the manuscript for publication in the journal.

Major remarks

1. I previously pointed out “In line with the previous point, how could capTEs be adapted to profile other type of TEs? There should be a more thorough discussion on the limitations and requirements (or future steps) that would allow capTEs to profile other type of TEs”

The authors discussed considerably the point I made before of using capTEs to detect other type of TEs. The discussion of this issue is an excellent addition to their work.

Later they mention that “Taking all detected TEs into account, we found no significant difference in the proportion of TE subfamilies between the TE loci identified by capTEs and those detected through total RNA-seq (Fig. 2d)”. What is the total number of TEs detected with total RNA-Seq? What fraction of those are detected by capTEs? Since capTEs was used on specific subfamilies, it would be useful to know how many TEs would be missed when compared to a regular RNA-Seq experiment.

2. In the analysis of “Locus-specific quantification of TE expression “, the authors expanded their comparison of the TE expression obtained with capTEs to that obtained with TETRanscripts and Telescope, with the latter being suggested by me and the other reviewer. This provided another layer for their analysis, in which they also expanded upon.

Similar to the point raised before, how do the proportions of total detected TEs vary compared to those detected by TETRanscripts and by Telescope? Of those detected with the tools, what is the proportion of young vs old TEs?

Addressing and discussing this point would further highlight potential advantages of capTEs over specific subfamilies of young TEs.

3. In “Characterizing the impacts of internal and inserted TEs on the transcriptome of cancer cells”, there are a few points that I don’t see described in detail.

How did the authors assess the overlaps between TEs and genes? What tool did they use? From what can be seen in the methods, this is not detailed.

In regards to this point, they also mention “We further analyzed TE transcripts that contained exonic Alu or L1 in Gencode annotations”. How was this assessed?

When they mention “host genes” (lines 609-610), what tool was used? Were TEs fully contained within exons consider? In the case of exon-overlapping TEs (as those described above), was there any threshold considered regarding the amount of overlap between the TE an exons?

Minor remarks

1. Line 578: “genome reference hg38” to “Human reference genome version hg38”
2. Line 600-601: “The oncogenes referred to the published list”. This should be moved to the beginning of the paragraph, and changed to “The list of oncogenes was obtained from a previously published work (reference) and used for the analyses described next”.
3. Figure 4g. The color of the box plots in the inset depicting expression are not described in the legend
4. Figure 5g. Although the author implemented some of the suggestions made in the first round of revisions, now it is unclear how they fit both “Relative No. of TE insertions” and “gene expression changes (Log2 fold change) in the same scale of the plot. This should be clarified in detail in the legend and methods.

Reviewer #2 (Remarks to the Author):

I am satisfied with the authors' efforts for the revision. All my questions have been properly addressed, and I have no further comments.

Reviewer #1 (Remarks to the Author):

In the revised version of their work, Li et al. improved upon several of the points raised in the first review round. Although most of their answers are satisfactory, there are still some points that need to be improved for recommendation of the manuscript for publication in the journal.

Response: Thanks for the valuable comments. We have carefully considered each comment. Below are our point-by-point responses. Modified text is in blue.

Major remarks

1. I previously pointed out “In line with the previous point, how could capTEs be adapted to profile other type of TEs? There should be a more thorough discussion on the limitations and requirements (or future steps) that would allow capTEs to profile other type of TEs”

The authors discussed considerably the point I made before of using capTEs to detect other type of TEs. The discussion of this issue is an excellent addition to their work.

Later they mention that “Taking all detected TEs into account, we found no significant difference in the proportion of TE subfamilies between the TE loci identified by capTEs and those detected through total RNA-seq (Fig. 2d)”. What is the total number of TEs detected with total RNA-Seq? What fraction of those are detected by capTEs? Since capTEs was used on specific subfamilies, it would be useful to know how many TEs would be missed when compared to a regular RNA-Seq experiment.

Response: Thanks for the valuable feedback. In this study, we designed gRNAs to target Alu and L1 elements. Using this gRNA pool, capTEs detected all branched subfamilies from Alu and L1 elements that were detected by total RNA-seq (performed on the Nanopore platform). This result has been included in the previous manuscript (Supplementary Table 2, page 5 lines 91-93). In all analyses, we considered Alu and L1 elements as the target TEs.

To assess the ability of our developed method to capture target TE loci, we compared Alu and L1 loci detected by capTEs with those from total RNA-seq. Our method detected 209,646 sites, which is five times more than the 30,189 identifications by total RNA-seq. Among the total target TE loci detected by these two methods, capTEs achieved a detection rate of 98.5%, leaving 3,256 loci undetected, while total RNA-seq only detected 14.2%. This further confirms the effectiveness of our method in detecting target TEs. We have included these additional results in the revised version (Fig. 2b, page 8 lines 143-151).

Considering that the gRNA pool we used in this study targets more than one subfamily of Alu and L1 elements, we investigated whether our method exhibited any bias toward some specific subfamilies within these target TEs compared to the regular total RNA-seq. To assess this, we analyzed the subfamily proportions in the target TE loci. In the previous manuscript, the description “Taking all detected TEs into account” was misleading. In the analysis of subfamily proportion, only the target TEs (Alu and L1) were counted. We have modified the text to improve the clarity (pages 8-9 lines 157-162) and provided the number of TE sites of each subfamily in Supplementary Table 3.

Regarding the analysis of young TE proportions, our method does not show any preference for young TEs compared to total RNA-seq. We detected five times more target TE loci than total RNA-seq, and a similar increase was observed in the detection of young TE loci. We attributed this result to the similar read lengths of these two methods which were both based on the Nanopore platform. This hypothesis was supported by the result that our method exhibited an approximately 6-fold enrichment in detecting young TEs when compared to the short-read NGS

data (Fig. 2d-e).

Furthermore, as we focused on testing our method for capturing target TEs in the “Enrichment of target TEs” section, we did not analyze other types of TEs beyond Alu and L1. We hope that these additional analyses and explanations have addressed the concerns raised.

Pages 8-9 (lines 139-166) Enrichment of target TEs: RNA sequencing is the most commonly used tool for the genome-wide analysis of TE expression. To assess the capability of our developed method for detecting target TEs, we generated 6 Gb of data using capTEs and total RNA-seq methods on the ONT platform and performed a comprehensive comparison between the two methods. The capTEs approach identified a total of 209,646 loci from the Alu and L1 elements, which was five times greater than the 30,189 identifications obtained from total RNA-seq (Fig. 2b). Considering all the Alu and L1 loci identified by both methods (212,902 loci), capTEs achieved a detection rate of 98.5%, leaving only 3,256 loci undetected (Fig. 2b). Likewise, capTEs outperformed total RNA-seq in identifying young TE sites (defined as those less than two million years old). It captured 97% of the total number of young TE loci detected by both methods in Alu and L1 elements, which was five times higher than the detection rate achieved by total RNA-seq (Fig. 2b). Moreover, capTEs exhibited higher coverage for the detected target TEs (Fig. 2c). Genomic DNA would interfere with the detection of transcribed TEs, especially when using targeted sequencing. To examine whether the efficient detection of target TEs using our method was a result of genomic DNA contamination, we included control samples that were not subjected to reverse transcription. In these samples, we did not detect TE signals, which strengthens the validity of our findings. To investigate whether capTEs exhibited any bias toward some specific subfamilies of Alu and L1 elements, we conducted a comparative analysis of TE subfamily proportions within the target TE loci identified by capTEs and total RNA-seq. Our analysis demonstrated no significant difference in the overall proportions of TE subfamilies between the two methods (Fig. 2d, Supplementary Table 3). The maximum enriched fold observed in the capTEs dataset (the ratio between the subfamily proportion of capTEs and total RNA-seq) was limited to 1.2 (Supplementary Table 3). These results indicate that our method effectively and modestly detects various subfamily members of Alu and L1 elements without displaying a substantial bias toward specific subfamilies.

Page 28 (lines 581-582) Methods: Young TEs were defined as those with an age of less than two million years.

Page 40 Fig 2b:

b, Venn diagram showing the overlap between the target TE and young TE loci detected by capTEs and total RNA-seq methods.

Page 13 Supplementary Materials, Supplementary Table 3:

Supplementary Table 3. Subfamily proportions within Alu and L1 elements detected by capTEs and total RNA-seq methods. The subfamily proportion refers to the percentage of a specific subfamily among all the Alu and L1 loci detected by the corresponding method. The enrichment fold was defined as the ratio of the subfamily proportion detected by capTEs to that detected by total RNA-seq.

Alu subfamilies					
Subfamily	capTEs detected loci	Total RNA-seq detected loci	Subfamily proportion (capTEs)	Subfamily proportion (total RNA-seq)	Enrichment fold
AluS	3221541	572830	61.50%	61.45%	1.0
AluY	812884	142934	15.52%	15.33%	1.2
AluJ	933656	164174	17.82%	17.61%	1.0
FAM	2816	430	0.05%	0.05%	1.0
FRAM	52299	8956	1.00%	0.96%	1.0
FLAM	215230	42910	4.11%	4.60%	0.9
L1 subfamilies					
Subfamily	capTEs detected loci	Total RNA-seq detected loci	Subfamily proportion (capTEs)	Subfamily proportion (total RNA-seq)	Enrichment fold
L1P	178616	26508	93.93%	92.17%	1.0
L1HS	11538	2252	6.07%	7.83%	0.8

2. In the analysis of “Locus-specific quantification of TE expression “, the authors expanded their comparison of the TE expression obtained with capTEs to that obtained with Tetranscripts and Telescope, with the latter being suggested by me and the other reviewer. This provided another layer for their analysis, in which they also expanded upon.

Similar to the point raised before, how do the proportions of total detected TEs vary compared

to those detected by Tetrascripts and by Telescope? Of those detected with the tools, what is the proportion of young vs old TEs?

Addressing and discussing this point would further highlight potential advantages of capTEs over specific subfamilies of young TEs.

Response: Thanks for the constructive comments. As suggested by the reviewer, we conducted an analysis to compare the subfamily proportions within the target TE loci identified by capTEs and NGS. We observed a substantially higher representation of evolutionarily young TE subfamilies, specifically Alu Y and L1Hs, in the capTEs results (Fig. 4d). Furthermore, the analysis of young TE loci revealed consistent results that the percentage of young TEs was six times higher in the target TE loci detected using capTEs than in those detected using NGS (Fig. 4e). These additional analyses demonstrate the advantage of our method over the NGS-based method in analyzing young TEs. We have incorporated these relevant results into the revised manuscript (page 14 lines 265-277).

In response to the comments regarding the proportions of total detected TEs compared to those detected by Tetrascripts and by Telescope, we would like to provide additional information regarding the detection and quantification of expressed TE sites based on NGS data. To evaluate the performance of Telescope and Tetrascripts, we analyzed the same NGS datasets and compared their outputted TE loci. However, the overlap between these two methods accounted for only 37% of the total quantified loci (as shown in figure A below). As is known, short-read sequencing faces challenges in TE analysis due to alignment issues, potentially leading to false positives, where some reads may be incorrectly assigned to nonexpressed TE loci. Telescope, tested as the best-performing bioinformatics tool among those designed specifically for locus-specific TE expression analysis (PMCID: PMC8769692), has demonstrated significant improvements over Tetrascripts in read assignments, effectively reducing false-positive rates (PMCID: PMC6786656). Consistently, the number of target TE loci quantified by Telescope was approximately 25% fewer than that quantified by Tetrascripts in our analysis. However, Telescope still has some unsolved false-positive issues (PMCID: PMC8769692). We found that 37% of the TE loci were quantified uniquely by Telescope when compared to Tetrascripts. Due to the uncertainty in the accuracy of expressed TE loci reported by Tetrascripts and Telescope, we refrained from using these loci based on NGS data as a control to assess the detection rate achieved by our method.

Moving on to the analysis of "Locus-specific quantification of TE expression", we first demonstrated the accuracy of our method in quantifying TE expression levels by employing TE cDNA standards. Considering that existing tools for TE expression analysis are predominantly based on short-read sequencing, we introduced NGS data to investigate the advantages provided by our long-read capTEs. We conducted a comparison of the proportion of young TEs and differentially expressed TE loci among the target TE loci detected by each method. These results consistently supported the superiority of our capTEs method over NGS-based methods in quantifying young TEs and detecting differentially expressed TE loci.

A, Venn diagram showing the overlap between the target TE loci quantified by Telescope and Tetranscripts.

Pages 13-15 (lines 260-286) Locus-specific quantification of TE expression: We next compared our TE expression results to those obtained from NGS data using two independent TE-dedicated bioinformatics tools, Tetranscripts and Telescope. Our analysis revealed that capTEs detected expression changes at 65,631 Alu and L1 loci between MDA-MB-231 and MCF 10A cells, while Tetranscripts and Telescope quantified 25,383 and 19,124 loci, respectively, based on NGS data (Fig. 4c). **When examining the subfamily proportions, we observed a substantially higher representation of evolutionarily young TE subfamilies, specifically Alu Y and L1Hs, within the target TE loci quantified by capTEs compared to NGS-based methods (Fig. 4d, Supplementary Table 4).** The proportions of Alu Y and L1Hs in capTEs-quantified target TEs were approximately 1.8 times and 17.6 times, respectively, of those observed in the NGS results (Fig. 4d, Supplementary Table 4). Moreover, our analysis of young TE loci revealed that their proportion within the Alu and L1 loci detected by capTEs was on average 5.8 times higher compared to NGS-based methods (5-fold higher than Tetranscripts and 6.5-fold higher than Telescope, Fig. 4e). These results highlight the superiority of capTEs over NGS in quantifying the expression of young TEs, which are known to exhibit higher sequence similarity than old TEs. We suspect that this advantage may be attributed to the long-read nature of our capTEs data. We further compared our quantification results to those reported by Telescope, as this tool was specifically designed for the locus-specific analysis of TE expression. Two pieces of evidence suggested that capTEs presented a higher sensitivity than NGS in detecting modest changes in the expression of individual TEs. First, **compared to NGS, capTEs identified more differentially expressed TEs at various evolutionary ages, and these differential TE sites accounted for a greater proportion of all quantified target TE sites (Fig. 4f, Supplementary Figure 6a).** Second, we observed lower expression changes for differential TEs detected only by capTEs in comparison to those revealed by both capTEs and NGS (Supplementary Figure 6b).

Page 42 Fig 4d-e:

d, Bar plots showing the subfamily proportions of target TEs detected by capTEs (blue), Telescope (orange) and Tetrascripts (gray). **e**, Bar plot showing the number of young TEs quantified with capTEs (blue), Telescope (orange) and Tetrascripts (gray). The percentages within the parentheses represent the proportion of young TEs among all the target TEs quantified by the respective method.

Page 13 Supplementary Materials, Supplementary Table 4:

Supplementary Table 4. Alu and L1 subfamilies quantified by capTEs and NGS methods. NGS data were analyzed using Telescope and Tetrascripts. The subfamily proportion refers to the proportion of a specific subfamily among all the detected TE loci. The enrichment fold was defined as the ratio of the subfamily proportion detected by capTEs to that detected by NGS method.

Alu subfamilies								
Subfamily	capTEs		NGS (Telescope)			NGS (Tetrascripts)		
	Detected site	Subfamily proportion	Detected site	Subfamily proportion	Enrichment fold	Detected site	Subfamily proportion	Enrichment fold
AluJ	10323	15.99%	4153	23.53%	0.68	5736	24.89%	0.64
AluS	42372	65.65%	11354	64.32%	1.02	14704	63.80%	1.03
AluY	10453	16.19%	1639	9.29%	1.74	2049	8.89%	1.82
FAM	20	0.03%	15	0.08%	0.36	24	0.10%	0.30
FLAM	1239	1.92%	410	2.32%	0.83	436	1.89%	1.01
FRAM	140	0.22%	81	0.46%	0.47	99	0.43%	0.50
L1 subfamilies								
Subfamily	capTEs		NGS (Telescope)			NGS (Tetrascripts)		
	Detected site	Subfamily proportion	Detected site	Subfamily proportion	Enrichment fold	Detected site	Subfamily proportion	Enrichment fold
L1HS	40	4.28%	4	0.28%	15.47	5	0.22%	19.74
L1P	894	95.72%	1441	99.72%	0.96	2300	99.78%	0.96

3. In “Characterizing the impacts of internal and inserted TEs on the transcriptome of cancer cells”, there are a few points that I don’t see described in detail. How did the authors assess the overlaps between TEs and genes? What tool did they use? From what can be seen in the methods, this is not detailed.

In regards to this point, they also mention “We further analyzed TE transcripts that contained exonic Alu or L1 in Gencode annotations”. How was this assessed?

When they mention “host genes” (lines 609-610), what tool was used? Were TEs fully contained within exons consider? In the case of exon-overlapping TEs (as those described above), was there any threshold considered regarding the amount of overlap between the TE an exons?

Response: Thank you for your valuable comments. To assess the overlaps between TEs and genes, we first utilized BEDTools intersect with the default parameters to find the intersection between TEs and transcripts. For each TE locus, we recorded transcripts with an overlap of one or more base pairs and considered them TE-containing transcripts. The genes to which these transcripts belong were defined as the host genes of the corresponding TE site. We have now included these details in the updated Methods (Pages 29-30 lines 615-623). Besides, we provided more information about the analyses conducted in “Characterizing the impacts of internal and inserted TEs on the transcriptome of cancer cells” section in the updated Methods (Page 32 lines 684-690, Page 30 lines 630-646).

Regarding the definition of TE host genes, we chose not to set a specific threshold for the overlap length or proportion due to several considerations. Alu and L1 elements have their own promoters and can initiate transcription, resulting in downstream sequences being present in the transcript. However, Alu and L1 in the human genome are not always intact copies, leading to variations in the overlap length with exons. Furthermore, TE elements can also participate in alternative splicing, leading to partial or complete inclusion of their sequences in transcripts. Considering these complexities, we decided not to impose a fixed threshold for the overlap length or proportion. Instead, we considered all transcript overlaps with TE elements. This enabled us to comprehensively capture the potential effects of TEs on the transcriptome.

Regarding the statement “We further analyzed TE transcripts that contained exonic Alu or L1 in Gencode annotations” in the “Enrichment target TEs” section, our intention was to assess the ability of our method to detect TEs of interest at the transcript level. To achieve this, we aligned reads containing Alu or L1 elements to the Gencode annotated transcripts and quantified the number of matched transcripts. This allowed us to determine whether our captured Alu and L1 reads were derived from more transcripts compared to total RNA-seq. We have reworded the text in the updated manuscript to provide clearer explanations.

Page 9 (lines 166-170) Enrichment of target TEs: We also assessed the ability of capTEs to detect target TEs at the transcript level. By aligning Alu- and L1-containing reads to the Gencode annotated transcripts, we identified 31,395 matched transcripts using capTEs, which is twice the number detected using total RNA-seq (Fig. 2e).

Page 28 (lines 577-579) Methods: Reads containing identifiable Alu or L1 elements, as determined by RepeatMasker, were considered as TE-containing reads.

Pages 29-30 (lines 615-629) Methods: To identify noncanonical transcripts containing target

TEs, we utilized BEDTools intersect with the default parameters to find the intersection between the genomic features of the target TEs identified by RepeatMasker in the human genome and the noncanonical transcripts. The transcripts with an overlap of one or more base pairs were considered noncanonical TE-containing transcripts. Likewise, for each TE locus, canonical transcripts with at least one base pair overlap with the analyzed TE were recorded and considered TE-containing transcripts. The genes to which these TE-containing transcripts belong were designated as TE-hosting genes for the corresponding TE sites.

To investigate the correlation between the production of noncanonical transcripts and the expression changes of TE-hosting genes, we counted the noncanonical transcripts under their compared reference genes as reported by GffCompare. The relative number of noncanonical transcripts for each gene, calculated as the difference in noncanonical transcript counts between cancer cells and matched normal cells, was used for further analysis.

Page 32 (lines 684-690) Methods: To evaluate the impact of noncanonical transcripts and canonical transcripts on the expression changes of TE-hosting genes, we measured their expression levels in cancer cells and matched normal cells. Next, we calculated the differences in expression levels between cancer cells and normal cells for both noncanonical and canonical transcripts. The contribution of each type of transcript was determined by taking the ratio of its difference value to the sum of the difference values of both types of transcripts.

Page 30 (lines 630-646) Methods: Autonomously transcribed TEs were defined as those that exhibited at least one base pair overlap with the TSSs of noncanonical transcripts fully contained within intronic and intergenic regions. In the case of intron retention, TEs overlapped noncanonical transcripts downstream of the TSSs, and the transcription pattern of the overlapped transcript was classified as intron retention. For all other expressed TEs that overlapped with canonical or noncanonical transcripts but did not fall into the above two scenarios, they were classified as undergoing passive transcription. The corresponding overlapped transcripts were then recorded under the respective transcription modes of the analyzed TEs based on the descriptions provided.

To assess the transcriptional contribution of individual overexpressed TE loci under specific transcription modes to the expression levels of their host genes, we quantified the total expression levels of the transcripts recorded under each particular mode of the analyzed TE. We then calculated the proportion of these transcript expression levels relative to the overall expression levels of the TE-hosting gene. This proportion was used as the transcriptional contribution in the analyzed transcription modes for a specific TE site.

Minor remarks

1. Line 578: “genome reference hg38” to “Human reference genome version hg38”

Response: Thanks for the reviewer’s suggestion. We have corrected it.

Page 29 (lines 649-651) Methods: PALMER was used to detect TEs that had not been annotated in the human reference genome version hg38 with the merged BAM files (--type ALU/LINE).

2. Line 600-601: “The oncogenes referred to the published list”. This should be moved to the beginning of the paragraph, and changed to “The list of oncogenes was obtained from a previously published work (reference) and used for the analyses described next”.

Response: Thanks for the reviewer’s suggestion. We have modified it accordingly.

Page 31 (lines 669-670) Methods: The list of oncogenes was obtained from a previously published work⁷ and used for the analyses described next.

3. Figure 4g. The color of the box plots in the inset depicting expression are not described in the legend

Response: We have added the description in the revised manuscript.

Page 43 (line 964-971) Fig 4. h, Genome browser view showing an example of measuring autonomous transcription levels of TEs at specific loci. The boxplot displays the expression levels of horizontally aligned transcripts. The colors, from bottom to top, indicate the transcripts where the analyzed TE was in autonomous (red), autonomous (orange) and passive (blue) transcription modes. The autonomous transcription level of this TE locus is represented by the total expression levels of the two transcripts labeled as "autonomous". The expression levels are indicated by normalized read counts.

4. Figure 5g. Although the author implemented some of the suggestions made in the first round of revisions, now it is unclear how they fit both “Relative No. of TE insertions” and “gene expression changes (Log2 fold change) in the same scale of the plot. This should be clarified in detail in the legend and methods.

Response: The relative number of TE insertions for a particular gene represents the difference in TE insertion counts between the cancer cells and their matched normal cells. In response to the reviewer's suggestion, we have now provided detailed explanations in the methods section and modified the colors used in the plot to clearly distinguish between the "Relative No. of TE insertions" and "gene expression changes (Log2 fold change)". Additionally, we would like to emphasize that Figure 5g displays the raw data of "Relative No. of TE insertions" and "gene expression changes (Log2 fold change)" without any alterations or adjustments to the original values. We hope the revisions have addressed the concerns effectively.

Page 42 Fig 5g:

Page 31 (lines 657-661) Methods: To explore the potential correlation between TE insertions and the expression levels of their inserted genes, we quantified the number of TE insertions and calculated the difference in TE insertion counts between the cancer cells and their matched normal cell for each TE-inserted gene. The resulting difference value was used as the relative number of TE insertions for further investigation.

Reviewer #2 (Remarks to the Author):

I am satisfied with the authors' efforts for the revision. All my questions have been properly addressed, and I have no further comments.

REVIEWERS' COMMENTS:

Reviewer #1 (Remarks to the Author):

The authors made a considerable effort in addressing all my previous commentaries, which indeed improved substantially their manuscript.

I'm satisfied with their latest version, and the paper can be recommended for publication now.

Referee comments

Reviewer #1 (Remarks to the Author):

The authors made a considerable effort in addressing all my previous commentaries, which indeed improved substantially their manuscript.

I'm satisfied with their latest version, and the paper can be recommended for publication now.